# Sparse Training via Boosting Pruning Plasticity with Neuroregeneration

**Shiwei Liu[1]\*, Tianlong Chen[2], Xiaohan Chen[2], Zahra Atashgahi[3], Lu Yin[1], Huanyu Kou[4],**
**Li Shen[5], Mykola Pechenizkiy[1,6], Zhangyang Wang[2], Decebal Constantin Mocanu[1,3]**
[1]Eindhoven University of Technology, [2]University of Texas at Austin
[3]University of Twente,[4]University of Leeds,[5]JD Explore Academy, [6]University of Jyväskylä
{s.liu3,l.yin,m.pechenizkiy}@tue.nl, {tianlong.chen,xiaohan.chen,atlaswang}@utexas.edu
{z.atashgahi,d.c.mocanu}@utwente.nl, {khydouble1,mathshenli}@gmail.com

## Abstract

Works on lottery ticket hypothesis (LTH) and single-shot network pruning (SNIP) have raised a lot of attention currently on post-training pruning (iterative magnitude pruning), and before-training pruning (pruning at initialization). The former method suffers from an extremely large computation cost and the latter usually struggles with insufficient performance. In comparison, during-training pruning, a class of pruning methods that simultaneously enjoys the training/inference efficiency and the comparable performance, temporarily, has been less explored. To better understand during-training pruning, we quantitatively study the effect of pruning throughout training from the perspective of **pruning plasticity** (the ability of the pruned networks to recover the original performance). Pruning plasticity can help explain several other empirical observations about neural network pruning in literature. We further find that pruning plasticity can be substantially improved by injecting a brain-inspired mechanism called **neuroregeneration**, i.e., to regenerate the same number of connections as pruned. We design a novel gradual magnitude pruning (GMP) method, named gradual pruning with zero-cost neuroregeneration (**GraNet**), that advances state of the art. Perhaps most impressively, its sparse-to-sparse version for the first time boosts the sparse-to-sparse training performance over various dense-to-sparse methods with ResNet-50 on ImageNet without extending the training time. We release all codes in https://github.com/Shiweiliuiiiiiii/GraNet.

## 1 Introduction

Neural network pruning is the most common technique to reduce the parameter count, storage requirements, and computational costs of modern neural network architectures. Recently, post-training pruning [49, 29, 18, 47, 10, 54, 74, 5, 57, 75] and before-training pruning [31, 30, 67, 63, 6, 11] have been two fast-rising fields, boosted by lottery tickets hypothesis (LTH) [10] and single-shot network pruning (SNIP) [31]. The process of post-training pruning typically involves fully pre-training a dense network as well as many cycles of retraining (either fine-tuning [18, 17, 39] or rewinding [12, 54]). As the training costs of the state-of-the-art models, e.g., GPT-3 [4] and FixEfficientNet-L2 [64] have exploded, this process can lead to a large amount of overhead cost.

Recently emerged methods for pruning at initialization significantly reduce the training cost by identifying a trainable sub-network before the main training process. While promising, the existing methods fail to match the performance achieved by the magnitude pruning after training [11].

---

\*Partial of this work have been done when Shiwei Liu worked as an intern at JD Explore Academy.

35th Conference on Neural Information Processing Systems (NeurIPS 2021).

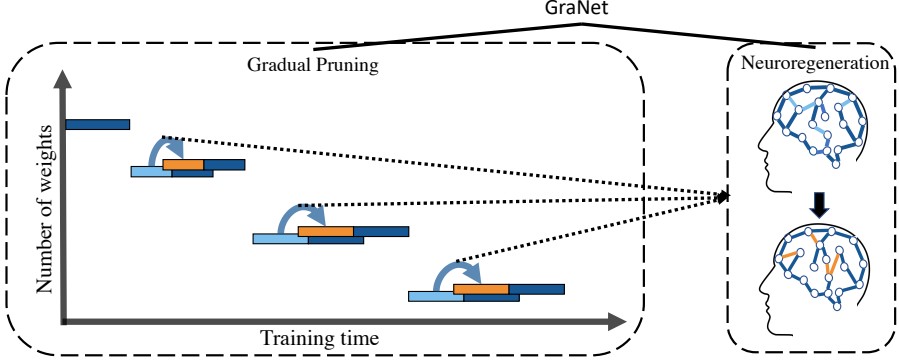

Figure 1: **Schematic view of GraNet. Left:** Gradual pruning starts with a sparse subnetwork and gradually prune the subnetwork to the target sparsity during training. **Right:** We perform zero-cost neuroregeneration after each gradual pruning step. Light blue blocks/lines refer to the "damaged" connections and orange blocks/lines refer to the regenerated new connections.

Compared with the above-mentioned two classes of pruning, during-training pruning is a class of methods that reap the acceleration benefits of sparsity early on the training and meanwhile achieve promising performance by consulting the information obtained during training. There are some works [77, 13, 33] attempting to gradually prune the network to the desired sparsity during training, while they mainly focus on the performance improvement. Up to now, the understanding of during-training pruning has been less explored due to its more complicated dynamical process, and the performance gap still exists between pruning during training and full dense training.

To better understand the effect of pruning during the optimization process (not at inference), we study the ability of the pruned models to recover the original performance after a short continued training with the *current* learning rate, which we call *pruning plasticity* (see Section 3.1 for a more formal definition). Inspired by the *neuroregeneration* mechanism in the nervous system where new neurons and connections are synthesized to recover the damage in the nervous system [26, 41, 73], we examine if allowing the pruned network to regenerate new connections can improve pruning plasticity, and hence contribute to pruning during training. We consequently propose a parameter-efficient method to regenerate new connections during the gradual pruning process. Different from the existing works for pruning understanding which mainly focus on dense-to-sparse training [42] (training a dense model and prune it to the target sparsity), we also consider sparse-to-sparse training (training a sparse model yet adaptively re-creating the sparsity pattern) which recently has received an upsurge of interest in machine learning [44, 3, 9, 48, 8, 37, 36].

In short, we have the following main findings during the course of the study:

**#1. Both pruning rate and learning rate matter for pruning plasticity.** When pruned with low pruning rates (e.g., 0.2), both dense-to-sparse training and sparse-to-sparse training can easily recover from pruning. On the contrary, if too many parameters are removed at one time, almost all models suffer from accuracy drops. This finding makes a connection to the success of the iterative magnitude pruning [10, 54, 5, 6, 65], where usually a pruning process with a small pruning rate (e.g., 0.2) needs to be iteratively repeated for good performance.

Pruning plasticity also gradually decreases as the learning rate drops. When pruning happens during the training phase with large learning rates, models can easily recover from pruning (up to a certain level). However, pruning plasticity drops significantly after the second learning rate decay, leading to a situation where the pruned networks can not recover with continued training. This finding helps to explain several observations (1) for gradual magnitude pruning (GMP), it is always optimal to end pruning before the second learning rate drop [77, 13]; (2) dynamic sparse training (DST) benefits from a monotonically decreasing pruning rate with cosine or linear update schedule [8, 9]; (3) rewinding techniques [12, 54] outperform fine-tuning as rewinding retrains subnetworks with the original learning rate schedule whereas fine-tuning often retrains with the smallest learning rate.

**#2. Neuroregeneration improves pruning plasticity.** Neuroregeneration [41, 73] refers to the regrowth or repair of nervous tissues, cells, or cell products. Conceptually, it involves synthesizing new neurons, glia, axons, myelin, or synapses, providing extra resources in the long term to replace

those damaged by the injury, and achieving a lasting functional recovery. Such mechanism is closely related to the brain plasticity [51], and we borrow this concept to developing a computational regime.

We show that, while regenerating the same number of connections as pruned, the pruning plasticity is observed to improve remarkably, indicating a more neuroplastic model being developed. However, it increases memory and computational overheads and seems to contradict the benefits of pruning-during-training. This however raises the question: *can we achieve efficient neuroregeneration during training with no extra costs*? We provide an affirmative answer to this question.

**#3. Pruning plasticity with neuroregeneration can be leveraged to substantially boost sparse training performance.** The above-mentioned findings of pruning plasticity can generalize to the final performance level under a full continued training to the end. Imitating the neuroregeneration behavior [41, 73], we propose a new sparse training method – gradual pruning with zero-cost neuroregeneration (GraNet), which is capable of performing regeneration without increasing the parameter count.

In experiments, GraNet establishes the new state-of-the-art performance bar for dense-to-sparse training and sparse-to-sparse training, respectively. Particularly, the latter for the first time boosts the sparse-to-sparse training performance over various dense-to-sparse methods by a large margin without extending the training time, with ResNet-50 on ImageNet. Besides the consistent performance improvement, we find the subnetworks that GraNet learns are more accurate than the ones learned by the existing gradual pruning method, providing explanations for the success of GraNet.

## 2  Related Work

**Post-Training Pruning.** Methods that yield a sparse neural network from a pre-trained network by pruning the unimportant weights or neurons, to the best of our knowledge, were proposed in [24] and [50]. After that, various pruning methods have emerged to provide increasingly efficient methods to identify sparse neural networks for inference. The pruning criterion includes weight magnitude [18, 10], gradient [61] Hessian [29, 19, 59], Taylor expansion [47, 46], etc. Low-rank decomposition [7, 23, 17, 71] are also used to induce structured sparsity in terms of channels or filters. Most of the above-mentioned pruning methods require many pruning and re-training cycles to achieve the desired performance.

**During-Training Pruning.** Instead of inheriting weights from a pre-trained model, some works attempt to discover well-performing sparse neural networks with one single training process.

Gradual Magnitude Pruning (GMP), introduced in [77] and studied further in [13], gradually sparsifies the neural network during the training process until the desired sparsity is reached. Besides, [40] and [68] are prior works that enforce the network to sparse during training via $L_0$ and $L_1$ regularization, respectively. [60, 34, 55, 70, 28] moved further by introducing trainable sparsity heuristics to learn the sparse masks and weights simultaneously. These methods are all classified as dense-to-sparse training as they start from a dense network.

Dynamic Sparse Training (DST) [44, 3, 48, 8, 9, 36, 35, 25] is another class of methods that prune models during training. The key factor of DST is that it starts from a random initialized sparse network and optimizes the sparse topology as well as the weights simultaneously during training (sparse-to-sparse training). Without an extended training time [37], sparse-to-sparse training usually falls short of dense-to-sparse training in terms of the prediction accuracy. For further details, see the survey of [43, 21].

**Before-Training Pruning.** Motivated by SNIP [31], many works [67, 63, 6] have emerged recently to explore the possibility of obtaining a trainable sparse neural network before the main training process. [11] demonstrates that the existing methods for pruning at initialization perform equally well when the unpruned weights are randomly shuffled, which reveals that what these methods discover is the layer-wise sparsity ratio, rather than the indispensable weight values and positions. Our analysis shows that both the mask positions and weight values are crucial for GraNet.

## 3  Methodology for Pruning Plasticity

The primary goal of this paper is to study the effect of pruning as well as neuroregeneration on neural networks during the standard training process. Therefore, we do not consider post-training pruning and before-training pruning. Below, we introduce in detail the definition of pruning plasticity and the experimental design that we used to study pruning plasticity.

## 3.1 Metrics

Let us denote $W_t \in \mathbb{R}^d$ as the weights of the network and $m_t \in \{0,1\}^d$ as the binary mask yielded from the pruning method at epoch $t$. Thus, the pruned network can be denoted as $W_t \odot m_t$. Let $T$ be the total number of epochs the model should be trained. Let $\text{CONTRAIN}^k(W_t \odot m_t, a)$ refers to the function that continues to train the pruned model for $k$ epochs with the learning rate schedule $a$.

**Definition of Pruning plasticity.** We define pruning plasticity as $t_{\text{CONTRAIN}^k(W_t \odot m_t, a_t)} - t_{\text{PRE}}$, where $t_{\text{PRE}}$ is the test accuracy measured before pruning and $t_{\text{CONTRAIN}^k(W_t \odot m_t, a_t)}$ is the test accuracy measured after $k$ epoch of continued training $\text{CONTRAIN}^k(W_t \odot m_t, a_t)$. Specifically, to better understand the effect of pruning on the current model status and to avoid the effect of learning rate decay, we fix the learning rate as the one when the model is pruned, i.e, $a_t$. This setting is also appealing to GMP [77, 13] and DST [44, 9, 48, 37] in which most of the pruned models are continually trained with the current learning rate for some time.

**Final performance gap.** Nevertheless, we also investigate the effect of pruning on the final performance, that is, continually training the pruned networks to the end with the remaining learning rate schedule $\text{CONTRAIN}^{T-t}(W_t \odot m_t, a_{[t+1:T]})$. In this case, we report $t_{\text{CONTRAIN}^{T-t}(W_t \odot m_t, a_{[t+1:T]})} - t_{\text{FINAL}}$, where $t_{\text{FINAL}}$ is the final test accuracy of the unpruned models.

## 3.2 Architectures and Datasets

We choose two commonly used architectures to study pruning plasticity, VGG-19 [58] with batch normalization on CIFAR-10 [27], and ResNet-20 [20] on CIFAR-10.

We share the summary of the networks, data, and hyperparameters of dense-to-sparse training in Table 1. We use standard implementations and hyperparameters available online, with the exception of the small batch size for the ResNet-50 on ImageNet due to the limited hardware resources ($2\times$ Tesla V100). All accuracies are in line with the baselines reported in the references [8, 11, 67, 9, 37].

Table 1: Summary of the architectures and hyperparameters we study in this paper.

| Model | Data | #Epoch | Batch Size | LR | LR Decay, Epoch | Weight Decay | Test Accuracy |
|---|---|---|---|---|---|---|---|
| ResNet-20 | CIFAR-10 | 160 | 128 | 0.1 ($\beta = 0.9$) | $10\times$, [80, 120] | 0.0005 | 92.41±0.04 |
| VGG-19 | CIFAR-10 | 160 | 128 | 0.1 ($\beta = 0.9$) | $10\times$, [80, 120] | 0.0005 | 93.85±0.05 |
| | CIFAR-100 | 160 | 128 | 0.1 ($\beta = 0.9$) | $10\times$, [80, 120] | 0.0005 | 73.43±0.08 |
| ResNet-50 | CIFAR-10 | 160 | 128 | 0.1 ($\beta = 0.9$) | $10\times$, [80, 120] | 0.0005 | 94.75±0.01 |
| | CIFAR-100 | 160 | 128 | 0.1 ($\beta = 0.9$) | $10\times$, [80, 120] | 0.0005 | 78.23±0.18 |
| | ImageNet | 100 | 64 | 0.1 ($\beta = 0.9$) | $10\times$, [30, 60, 90] | 0.0004 | 76.80±0.09 |

## 3.3 How to Prune, and How to Regenerate

**Structured and Unstructured Pruning.** We consider unstructured and structured pruning in this paper. Structured pruning prunes weights in groups, or removes the entire neurons, convolutional filters, or channels, enabling acceleration with the off-the-shelf hardware. In particular, we choose the filter pruning method used in Li et al. [32]. Unstructured sparsity is a more promising direction not only due to its outstanding performance at extreme sparsities but the increasing support for sparse operation in the practical hardware [35, 14, 52, 76, 22]. For example, Liu et al. [35] illustrated for the first time the true potential of DST, demonstrating significant training/inference efficiency improvement over the dense training. Different from prior conventions [77, 13, 33, 2] where values of the pruned weights are kept, we set the pruned weights to zero to eliminate the historical information for all implementations in this paper.

**Magnitude pruning.** We prune the weights with the smallest magnitude, as it has evolved as the standard method when pruning happens during training, e.g., GMP [77, 13] and DST [44, 9, 37]. We are also aware of other pruning criteria including but not limited to Hessian [29, 19, 59], Taylor expansion [47, 46], connection sensitivity [31], Gradient Flow [67], Neural Tangent Kernel [38, 16].

**One-shot pruning.** To isolate the pruning effect at different training stages and to avoid the interaction between two iterations of pruning, we focus on one-shot pruning. Please note that iterative pruning can also be generalized in our setting, as our experimental design includes neural networks trained at various sparsities and each of them is further pruned with various pruning rates.

**Layer-wise pruning and global pruning.** We study both the layer-wise magnitude pruning and global magnitude pruning for pruning plasticity. Global magnitude pruning prunes different layers

together and leads to non-uniform sparsity distributions; layer-wise pruning operates layer by layer, resulting in uniform distributions.

**Gradient-based regeneration.** The simplest regeneration scheme is to randomly activate new connections [3, 44]. However, it would take a lot of time for random regeneration to discover the important connections, especially for the very extreme sparsities. Alternatively, gradients, including those for the connections with zero weights, provide good indicators for the connection importance. For this reason, we focus on gradient-based regeneration proposed in Rigged Lottery ( RigL) [9], i.e., regenerating the same number of connections as pruned with the largest gradient magnitude.

### 3.4 Experimental Results

We study pruning plasticity during training with/without regeneration, for both dense training and sparse training. We report the results of ResNet-20 on CIFAR-10 with unstructured global pruning in the main body of the paper. The rest of the experiments are given in Appendix A. Unless otherwise stated, results are qualitatively similar across all networks. Concretely, we first pre-train networks at four sparsity levels, including 0, 0.5, 0.9, and 0.98. The sparse neural networks are trained with uniform distribution (i.e., all layers have the same sparsity). We further choose four pruning rates, e.g., 0.2, 0.5, 0.9, and 0.98, to measure the corresponding pruning plasticity of the pre-trained networks.

**Pruning plasticity.** We continue to train the pruned model for 30 epochs and report pruning plasticity in Figure 2. Overall, the learning rate schedule, the pruning rate, and the sparsity of the original models all have a big impact on pruning plasticity. Pruning plasticity decreases as the learning rate decays for all models with different sparsity levels. The models trained with a large learning rate 0.1 can easily recover, or exceed the original performance except for the extremely large pruning rate 0.98. However, the models obtained during the later training phases can recover only with the mild pruning rate choices, e.g., 0.2 (orange lines) and 0.5 (green lines).

We next demonstrate the effect of connection regeneration on pruning plasticity in the bottom row of Figure 2. It is clear to see that connection regeneration significantly improves pruning plasticity of all the cases, especially for the models that are over-pruned (purple lines). Still, even with connection regeneration, pruning plasticity suffers from performance degradation when pruning occurs after the learning rate drops.

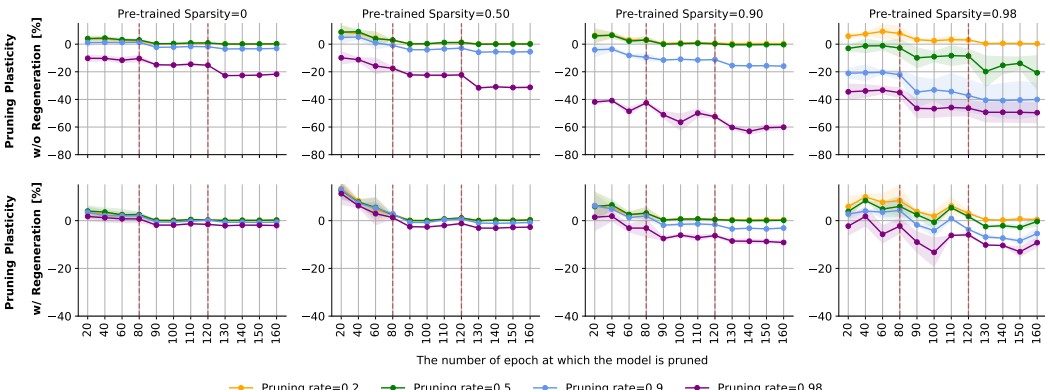

Figure 2: **Unstructured Pruning:** Pruning plasticity (see Section 3.1 for definition) under a 30-epoch continued training with and without connection regeneration for ResNet-20 on CIFAR-10. The vertical red lines refer to the points when the learning rate is decayed. "Pre-trained Sparsity" refers to the original sparsity of the pre-trained networks before pruning. The pruning method is the magnitude global pruning.

**Final performance gap.** Compared with the current model status, people might be more interested in the effect of pruning on the final performance. We further measure the performance gap between the original test accuracy of the unpruned models and the final test accuracy of the pruned model under a full continued training $\text{CONTRAIN}^{T-t}(W_t \odot m_t, a_{[t+1:T]})$ in Figure 3.

We observe that, in this case, large learning rates do not enjoy large performance improvement, but still, the performance gap increases as the learning rate drops. It is reasonable to conjecture that the accuracy improvement of pruning plasticity with the large learning rate, 0.1, is due to the

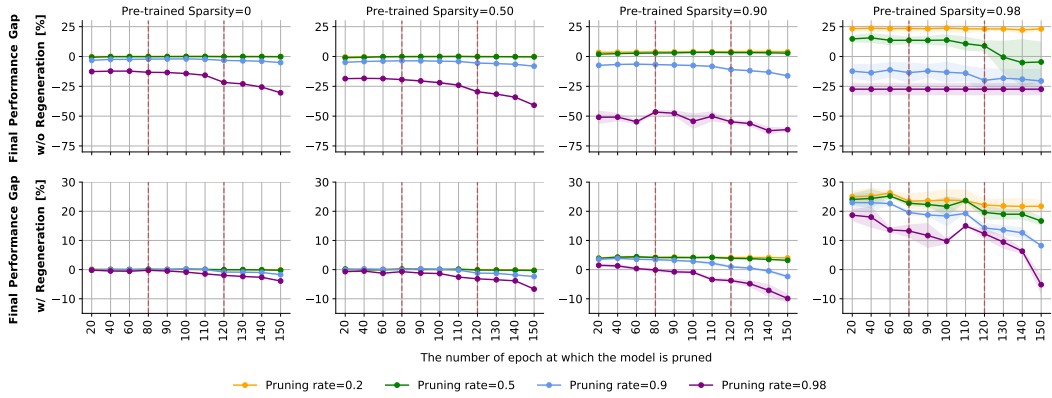

Figure 3: **Unstructured Pruning:** Final performance gap between the unpruned models and the pruned models for ResNet-20 on CIFAR-10. The vertical red lines refer to the points when the learning rate is decayed. "Pre-trained Sparsity" refers to the original sparsity of the pre-trained networks before pruning. The pruning method is the magnitude global pruning.

unconverged performance during the early phase of training. Besides, it is surprising to find that the final performance of extreme sparse networks (e.g., the third column and the fourth column) significantly benefits from mild pruning. Again, the ability of the pruned model to recover from pruning remarkably improves after regenerating the connections back.

## 4    Gradual Pruning with Zero-Cost Neuroregeneration

So far, we have known that regenerating the important connections to the pruned models during training substantially improves pruning plasticity as well as the final performance. However, naively regenerating extra connections increases the parameter count and conflicts with the motivation of gradual pruning.

Inspired by the mechanism of neuroregeneration in the nervous system, we propose a novel sparse training method which we call gradual pruning with zero-cost neuroregeneration (GraNet). GraNet consults the information produced throughout training and regenerates important connections during training in a parameter-efficient fashion. See Appendix B.1 for the pseudocode of GraNet. We introduce the main components of GraNet below.

### 4.1    Gradual Pruning

We follow the gradual pruning scheme used in [77] and gradually sparsifies the dense network to the target sparsity level over $n$ pruning iterations. Let us define $s_i$ is the initial sparsity, $s_f$ is the target sparsity, $t_0$ is is the starting epoch of gradual pruning, $t_f$ is the end epoch of gradual pruning, and $\Delta t$ is the pruning frequency. The pruning rate of each pruning iteration is:

$$s_t = s_f + (s_i - s_f) \left( 1 - \frac{t - t_0}{n\Delta t} \right)^3 , \ t \in \{t_0, t_0 + \Delta t, ..., t_0 + n\Delta t\} . \qquad (1)$$

We choose global pruning for our method as it generally achieves better performance than uniform pruning. We also report the performance of the uniform sparsity as used in [13] in Appendix C.3.

The conventional gradual pruning methods [77, 13] change the mask (not the weight values) to fulfill the pruning operation, so that the pruned connections have the possibility to be reactivated in the later training phases. Despite this, since the weights of the pruned connections are not updated, they have a small chance to receive sufficient updates to exceed the pruning threshold. This hinders the regeneration of the important connections.

### 4.2    Zero-Cost Neuroregeneration

The main difference between GraNet and the conventional GMP methods [77, 13] is the Zero-Cost Neuroregeneration. Imitating the neuroregeneration of the peripheral nervous system [41, 73] where new neurons and connections are synthesized to replace the damaged ones, we first detect and eliminate the "damaged" connections, and then regenerate the same number of new connections. By doing this, we can achieve connection regeneration without increasing the number of connections.

Concretely, we identify the "damaged" connections as the ones with the smallest weight magnitudes. Small magnitude indicates that either the weight's gradient is small or a large number of oscillations occur to the gradient direction. Therefore, these weights have a small contribution to the training loss and can be removed. Again, we use the gradient as the importance score for regeneration, same as the regrow method as used in RigL [9].

*Why we call it "Zero-Cost Neuroregeneration"?* In addition to not increasing the connection (parameter) count, the backward pass of our method is sparse most of the time even though our regeneration utilizes the dense gradient to identify the important connections. We perform neuroregeneration immediately after each gradual pruning step, meaning that the regeneration occurs only once every several thousand iterations. The extra overhead to calculate the dense gradient can be amortized compared with the whole training costs. Compared with the methods [33, 69] that require updating all the weights in the backward pass, our method is much more training efficient, as around 2/3 of the training FLOPs is owing to the backward pass [9, 72].

Let us denote $r$ as the ratio of the number of the regenerated connections to the total number of connections; $W$ is the network weight. We first remove $r$ proportion of "damaged" weights with the smallest magnitude by:

$$W' = \text{TopK}\left(|W|,\ 1 - r\right). \tag{2}$$

Here $\text{TopK}(v, k)$ returns the weight tensor retaining the top $k$-proportion of elements from $v$. Immediately after that, we regenerate $r$ proportion of new connections based on the gradient magnitude:

$$W = W' + \text{TopK}\left(|\mathbf{g}_{i \notin W'}|,\ r\right), \tag{3}$$

where $|\mathbf{g}_{i \notin W'}|$ are the gradient magnitude of the zero weights. We perform Zero-Cost Neuroregeneration layer by layer from the beginning of the training to the end.

GraNet can naturally generalize to the dense-to-sparse training scenario and the sparse-to-sparse training scenario by setting the initial sparsity level $s_i = 0$ and $s_i > 0$ in Eq. (1), respectively. For simplicity, we set $s_i = 0.5$, $t_0 = 0$, and $t_f$ as the epoch when performing the first learning rate decay for the sparse-to-sparse training. Different from the existing sparse-to-sparse training methods, i.e., SET [44], RigL [9], and ITOP [37], in which the sparsity is fixed throughout training, GraNet starts from a denser yet still sparse model and gradually prunes the sparse model to the desired sparsity. Although starting with more parameters, the global pruning technique of gradual pruning helps GraNet quickly evolve to a better sparsity distribution than RigL with lower feedforward FLOPs and higher test accuracy. What's more, GraNet sparsifies all layers including the first convolutional layer and the last fully-connected layer.

### 4.3 Experimental Results

We conduct various experiments to evaluate the effectiveness of GraNet. We compare GraNet with various dense-to-sparse methods and sparse-to-sparse methods. The results of Rigged Lottery (RigL) and GMP with CIFAR-10/100 were reproduced by our implementation with PyTorch so that the only difference between GraNet and GMP is the Zero-Cost Neuroregeneration. For each model, we divide the results into three groups from top to bottom: pruning at initialization, dynamic sparse training and dense-to-sparse methods. See Appendix B for more implementation details used in the experiments. GraNet ($s_i = 0.5$) refers to the sparse-to-sparse version and the and GraNet ($s_i = 0$) refers to the dense-to-sparse version.

**CIFAR-10/100.** The results of CIFAR-10/100 are shared in Table 2. We can observe that performance differences among different methods on CIFAR-10 are generally small, but still, GraNet ($s_i = 0$) consistently improves the performance over GMP except for the sparsity 95%, and achieves the highest accuracy in 4 out of 6 cases. In terms of the more complex data CIFAR-100, the performance differences between the during-training pruning methods and before-training pruning methods are much larger. GraNet ($s_i = 0$) again consistently outperforms GMP with all sparsities, highlighting the benefits of Zero-Cost Neuroregeneration. It is maybe more interesting that GraNet ($s_i = 0$) even outperforms the post-training method, subdifferential inclusion for sparsity (SIS), by a large margin.

In terms of sparse-to-sparse training, our proposed GraNet ($s_i = 0.5$) has a dominant performance over other methods. Especially at the very extreme sparsity 0.98, our method outperforms RigL by 1.40% and 2.22% with VGG-19 on CIFAR-10 and CIFAR-100, respectively.

**ImageNet.** Due to the small data size, the experiments with CIFAR-10/100 may not be sufficient to draw a solid conclusion. We further evaluate our method with ResNet-50 on ImageNet in Table 3.

Table 2: Test accuracy of pruned VGG-19 and ResNet-50 on CIFAR-10/100. We mark the best sparse-to-sparse training results in blue and the best dense-to-sparse training results in bold. The results reported with (mean ± std) are run with three different random seeds by us. The rest are obtained from [66] and [67]. Note that the accuracy of RigL is higher than the ones reported in [66], as we choose a large update interval following the In-Time Over-Parameterization strategy [37]. $s_i$ refers to the initial sparsity of GraNet.

| Dataset | CIFAR-10 | | | CIFAR-100 | | |
|---|---|---|---|---|---|---|
| Pruning ratio | 90% | 95% | 98% | 90% | 95% | 98% |
| **VGG-19** (Dense) | 93.85±0.05 | - | - | 73.43±0.08 | - | - |
| SNIP [31] | 93.63 | 93.43 | 92.05 | 72.84 | 71.83 | 58.46 |
| GraSP [67] | 93.30 | 93.04 | 92.19 | 71.95 | 71.23 | 68.90 |
| SynFlow [63] | 93.35 | 93.45 | 92.24 | 71.77 | 71.72 | 70.94 |
| Deep-R [3] | 90.81 | 89.59 | 86.77 | 66.83 | 63.46 | 59.58 |
| SET [44] | 92.46 | 91.73 | 89.18 | 72.36 | 69.81 | 65.94 |
| RigL [9] | 93.38±0.11 | 93.06±0.09 | 91.98±0.09 | 73.13±0.28 | 72.14±0.15 | 69.82±0.09 |
| GraNet ($s_i = 0.5$) (ours) | **93.73±0.08** | **93.66±0.07** | **93.38±0.15** | **73.30±0.13** | **73.18±0.31** | **72.04±0.13** |
| STR [28] | 93.73 | 93.27 | 92.21 | 71.93 | 71.14 | 69.89 |
| SIS [66] | **93.99** | 93.31 | 93.16 | 72.06 | 71.85 | 71.17 |
| GMP [13] | 93.59±0.10 | 93.58±0.07 | 93.52±0.03 | 73.10±0.12 | 72.30±0.15 | 72.07±0.37 |
| GraNet ($s_i = 0$) (ours) | 93.80±0.10 | **93.72±0.11** | **93.63±0.08** | **73.74±0.30** | **73.10±0.04** | **72.35±0.26** |
| **ResNet-50** (Dense) | 94.75±0.01 | - | - | 78.23±0.18 | - | - |
| SNIP [31] | 92.65 | 90.86 | 87.21 | 73.14 | 69.25 | 58.43 |
| GraSP [67] | 92.47 | 91.32 | 88.77 | 73.28 | 70.29 | 62.12 |
| SynFlow [63] | 92.49 | 91.22 | 88.82 | 73.37 | 70.37 | 62.17 |
| RigL [9] | 94.45±0.43 | 93.86±0.25 | 93.26±0.22 | 76.50±0.33 | 76.03±0.34 | 75.06±0.27 |
| GraNet ($s_i = 0.5$) (ours) | **94.64±0.27** | **94.38±0.28** | **94.01±0.23** | **77.89±0.33** | **77.16±0.52** | **77.14±0.45** |
| STR [28] | 92.59 | 91.35 | 88.75 | 73.45 | 70.45 | 62.34 |
| SIS [66] | 92.81 | 91.69 | 90.11 | 73.81 | 70.62 | 62.75 |
| GMP [13] | 94.34±0.09 | **94.52±0.08** | 94.19±0.04 | 76.91±0.23 | 76.42±0.51 | 75.58±0.20 |
| GraNet ($s_i = 0$) (ours) | **94.49±0.08** | 94.44±0.01 | **94.34±0.17** | **77.29±0.45** | **76.71±0.26** | **76.10±0.20** |

Table 3: Test accuracy of pruned ResNet-50 on ImageNet dataset. The best results of DST methods are marked as blue and the best results of pruning during training methods are marked in bold. The training/test FLOPs are normalized with the FLOPs of a dense model. $s_i$ refers to the initial sparsity of GraNet.

| Method | Top-1 Accuracy | FLOPs (Train) | FLOPs (Test) | TOP-1 Accuracy | FLOPs (Train) | FLOPs (Test) |
|---|---|---|---|---|---|---|
| Dense | 76.8±0.09 | 1x (3.2e18) | 1x (8.2e9) | 76.8±0.09 | 1x (3.2e18) | 1x (8.2e9) |
| Pruning ratio | | 80% | | | 90% | |
| Static (ERK) | 72.1±0.04 | 0.42× | 0.42× | 67.7±0.12 | 0.24× | 0.24× |
| Small-Dense | 72.1±0.06 | 0.23× | 0.23× | 67.2±0.12 | 0.10× | 0.10× |
| SNIP [31] | 72.0±0.06 | 0.23× | 0.23× | 67.2±0.12 | 0.10× | 0.10× |
| SET [44] | 72.9±0.39 | 0.23× | 0.23× | 69.6±0.23 | 0.10× | 0.10× |
| DSR[48] | 73.3 | 0.40× | 0.40× | 71.6 | 0.30× | 0.30× |
| RigL (ERK) [9] | 75.1±0.05 | 0.42× | 0.42× | 73.0±0.04 | 0.25× | 0.24× |
| SNFS (ERK) [8] | 75.2±0.11 | 0.61× | 0.42× | 72.9±0.06 | 0.50× | 0.24× |
| GraNet ($s_i = 0.5$) (ours) | **76.0** | 0.37× | 0.35× | **74.5** | 0.25× | 0.20× |
| STR [28] | **76.1** | n/a | 0.17× | 74.0 | n/a | 0.08× |
| DPF [33] | 75.1 | 0.71× | 0.23× | n/a | n/a | n/a |
| GMP [13] | 75.6 | 0.56× | 0.23× | 73.9 | 0.51× | 0.10× |
| GraNet ($s_i = 0$) (ours) | 75.8 | 0.34× | 0.28× | **74.2** | 0.23× | 0.16× |

We only run this experiment once due to the limited resources. We set $t_0 = 0$ and $t_f = 30$ for both GraNet ($s_i = 0$) and GraNet ($s_i = 0.5$) on ImageNet. Again, GraNet ($s_i = 0$) outperforms GMP consistently with only half training FLOPs and achieves the highest accuracy among all the dense-to-sparse methods at sparsity of 0.9. Surprisingly, GraNet ($s_i = 0.5$) significantly boosts the sparse-to-sparse training performance, even over the dense-to-sparse training. Concretely, GraNet

($s_i = 0.5$) outperforms RigL by 0.9% and 1.5% at sparsity 0.8 and 0.9, respectively. To the best of our knowledge, this is the first time in the literature that sparse-to-sparse training reaches a test accuracy of 76% with ResNet-50 on ImageNet at sparsity 0.8, without extension of training time. It is reasonable for GraNet ($s_i = 0.5$) to achieve better accuracy than RigL, since the denser models at the beginning help GraNet explore more the parameter space. According to the In-Time Over-Parameterization hypothesis [37], the performance of sparse training methods is highly correlated with the total number of parameters that the sparse model has visited.

We further report the training/inference FLOPs required by all pruning methods. Compared with other dense-to-sparse methods, the final networks learned by GraNet ($s_i = 0$) require more FLOPs to test, whereas the overall training FLOPs required by GraNet ($s_i = 0$) are smaller than others. Even though starting from a denser model, GraNet ($s_i = 0.5$) requires less training and inference FLOPs than the state-of-the-art method, i.e., RigL. The sparsity budgets for 0.9 sparse ResNet-50 on ImageNet-1K learned by our methods are reported in Appendix D. We also report how FLOPs of the pruned ResNet-50 evolve during the course of training in Appendix E.

### 4.4 Effect of the Initial Sparsity

As we mentioned earlier, the denser initial network is the key factor in the success of GraNet. We conducted experiments to study the effect of the initial sparsity on GraNet with ResNet-50 on ImageNet. The initial sparsity is chosen from [0.0, 0.5, 0.6, 0.7, 0.8, 0.9] and the final sparsity is fixed as 0.9. The results are shared in Table 4. We can see the training FLOPs of GraNet are quite robust to the initial sparsity. Surprisingly yet reasonably, it seems that the the smaller the initial sparsity is (up to 0.5), the better final sparsity distribution GraNet finds, with higher test accuracy and fewer feedforward FLOPs. The lower feedforward FLOPs of the final network perfectly balance the overhead caused by the denser initial network.

Table 4: Effect of the initial sparsity on GraNet with ResNet-50 on ImageNet. The training/test FLOPs are normalized with the FLOPs of a dense model.

| Method | $s_i$ | $s_f$ | Top-1 [%] Accuracy | FLOPs (Train) | FLOPs (Test) |
|---|---|---|---|---|---|
| GraNet | 0.0 | 0.9 | 74.2 | 0.23× | 0.16× |
| GraNet | 0.5 | 0.9 | 74.5 | 0.25× | 0.20× |
| GraNet | 0.6 | 0.9 | 74.4 | 0.25× | 0.22× |
| GraNet | 0.7 | 0.9 | 74.2 | 0.24× | 0.22× |
| GraNet | 0.8 | 0.9 | 74.1 | 0.25× | 0.24× |
| RigL | 0.9 | 0.9 | 73.0 | 0.25× | 0.24× |

### 4.5 Performance of GraNet at Extreme Sparsities

In this section, we share the results of GraNet and RigL at extreme sparsities. The initial sparsity is set as 0.5. When the final sparsity is relatively smaller (e.g., 0.8, 0.9), GraNet requires a lower (or the same) number of training FLOPs than RigL, whereas GraNet requires more training FLOPs than RigL when the final sparsity is extremely high (e.g., 0.95, 0.965). This makes sense since when the sparsity is extremely high, the saved FLOPs count of the distribution discovered by GraNet is too small to amortize the overhead caused by denser initial models. Yet, the increased number of training FLOPs of GraNet leads to substantial accuracy improvement (> 2%) over RigL. The efficiency of GraNet ($s_i = 0.5$) comes from two important technical differences compared with RigL: (1) better final sparse distribution discovered by global pruning; (2) a shorter period of gradual pruning time

Table 5: Comparison between GraNet and RigL at extreme sparsities with ResNet-50 on ImageNet. The training/test FLOPs are normalized with the FLOPs of a dense model.

| Method | $s_i$ | $s_f$ | Top-1 [%] Accuracy | FLOPs (Train) | FLOPs (Test) |
|---|---|---|---|---|---|
| RigL | 0.8 | 0.8 | 75.1 | 0.42× | 0.42× |
| GraNet | 0.5 | 0.8 | 76.0 | 0.37× | 0.35× |
| RigL | 0.9 | 0.9 | 73.0 | 0.25× | 0.24× |
| GraNet | 0.5 | 0.9 | 74.5 | 0.25× | 0.20× |
| RigL | 0.95 | 0.95 | 69.7 | 0.12× | 0.12× |
| GraNet | 0.5 | 0.95 | 72.3 | 0.17× | 0.12× |
| RigL | 0.965 | 0.965 | 67.2 | 0.11× | 0.11× |
| GraNet | 0.5 | 0.965 | 70.5 | 0.15× | 0.09× |

(the first 30 epochs for ResNet-50 on ImageNet). Although starting with more parameters, the global pruning enables GraNet to quickly (first 30 epochs) evolve to a better sparsity distribution with lower test FLOPs than ERK. After 30 epochs of gradual pruning, the network continues to be trained with this better distribution for 70 epochs, so that the overhead in the early training phase with larger training FLOPs is amortized by the later and longer training phase with fewer training FLOPs.

### 4.6 Ablation Study of Random Reinitialization

Next, we ask whether what GraNet learned are the specific sparse connectivity or the sparse connectivity together with the weight values. We randomly reinitialize the pruned network with the

same mask and retrain it. The results are given in Figure 4. The performance of the reinitialized networks falls significantly short of the performance achieved by GraNet ($s_i = 0$), indicating that what was learned by GraNet is the sparse connectivity together with the weight values. Besides, we find that the retraining performance of GraNet is higher than GMP. This further confirms that Zero-Cost Neuroregeneration helps the gradual pruning find more accurate mask positions.

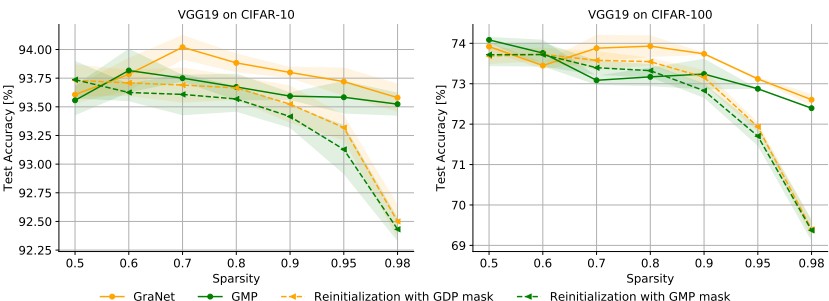

Figure 4: Reinitialization ablation on subnetworks discovered by GMP and GraNet ($s_i = 0$).

## 4.7   Comparison between Re-training and Extended Training

In this section, we study if re-training techniques can further improve the performance of the subnetworks discovered by GraNet. The authors of Lottery Ticket Hypothesis (LTH) [10] introduced a retraining technique, even if they did not evaluate it as such, where the subnetworks discovered by iterative magnitude pruning can be re-trained in isolation to full accuracy with the original initializations. Later on, learning rate rewinding (LRR) [54] was proposed further to improve the re-training performance by only rewinding the learning rate. Since GraNet also utilizes magnitude pruning to discover subnetworks, it is natural to test if these re-training techniques can bring benefits to GraNet. As shown in Table 6, both re-training techniques do not bring benefits to GraNet. Instead of re-training the subnetworks, we find that simply extending the training time significantly boosts the performance of GraNet with similar computational costs.

Table 6: Effects of LTH and LRR on the subnetworks learned by GraNet. Methods with $_2\times$ refer to extending the training steps by 2 times. The results are reported with top-1 test accuracy [%].

| Dataset | CIFAR-10 | | | CIFAR-100 | | |
|---|---|---|---|---|---|---|
| Pruning ratio | 90% | 95% | 98% | 90% | 95% | 98% |
| **VGG-19** (Dense) | 93.85±0.05 | - | - | 73.43±0.08 | - | - |
| GraNet ($s_i = 0$) | 93.80±0.10 | 93.72±0.11 | 93.63±0.08 | 73.74±0.30 | 73.10±0.04 | 72.35±0.26 |
| + Lottery Ticket Hypothesis | 93.63±0.04 | 93.29±0.05 | 92.46±0.08 | 72.97±0.25 | 71.76±0.22 | 69.28±0.36 |
| + Learning Rate Rewinding | 93.84±0.14 | 93.72±0.06 | 93.53±0.04 | 73.71±0.08 | 73.24±0.24 | 72.50±0.26 |
| GraNet$_{2\times}$ ($s_i = 0$) | **94.17±0.03** | **93.98±0.07** | **93.94±0.11** | **74.80±0.29** | **73.65±0.32** | **73.63±0.05** |
| **ResNet-50** (Dense) | 94.75±0.01 | - | - | 78.23±0.18 | - | - |
| GraNet ($s_i = 0$) | 94.49±0.08 | 94.44±0.01 | 94.34±0.17 | 77.29±0.45 | 76.71±0.26 | 76.10±0.20 |
| + Lottery Ticket Hypothesis | 93.96±0.10 | 93.70±0.15 | 92.94±0.14 | 75.74±0.19 | 74.31±0.10 | 71.99±0.08 |
| + Learning Rate Rewinding | 94.55±0.13 | 94.39±0.13 | 94.20±0.25 | 77.40±0.14 | 76.90±0.19 | 75.75±0.25 |
| GraNet$_{2\times}$ ($s_i = 0$) | **95.09±0.15** | **94.84±0.11** | **94.69±0.24** | **78.18±0.20** | **78.17±0.20** | **77.15±0.29** |

## 5   Conclusion, and Reflection of Broader Impacts

In this paper, we re-emphasize the merit of during-training pruning. Compared with the recently proposed works, i.e., LTH and SNIP, during-training pruning is an efficient yet performant class of pruning methods that have received much less attention. We quantitatively study pruning during training from the perspective of pruning plasticity. Inspired by the findings from pruning plasticity and the mechanism of neuroregeneration in the nervous system, we further proposed a novel sparse training method, GraNet, that performs the cost-free connection regeneration during training. GraNet advances the state of the art in both dense-to-sparse training and sparse-to-sparse training.

Our paper re-emphasizes the great potential of during-training pruning in reducing the training/inference resources required by ML models without sacrificing accuracy. It has a significant environmental impact on reducing the energy cost of the ML models and $CO_2$ emissions [1, 53, 15, 56, 62].

## 6 Acknowledgement

This project is partially financed by the Dutch Research Council (NWO). We thank the reviewers for the constructive comments and questions, which improved the quality of our paper.

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
