# A  Remaining Experimental Results of Pruning Plasticity

We also studied pruning plasticity on structured pruning. In particular, we choose the filter pruning method used in Li et al. [32]. The pruning criterion is the absolute weight sum of each nonzero filter and the regeneration criterion is the absolute gradient sum of each zero filter. We first pre-train four sets of neural networks from scratch with various structured sparsity, including 0, 0.10, 0.50, and 0.70, noted as "Pre-trained Sparsity" in the figure title. To measure the plasticity of these pre-trained models, we choose four different pruning rates noted as "Pruning rate" to remove filters from these pre-trained models. The results of ResNet-20 and VGG-19 are shown as below.

## A.1  ResNet-20 on CIFAR-10 with Structured Filter Pruning

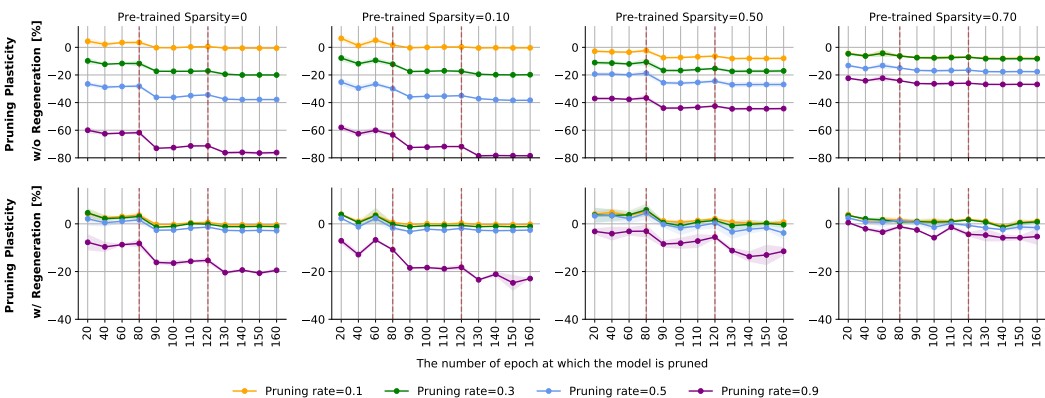

Figure 5: **Structured Pruning:** Pruning plasticity of under a 30-epochs continued training with and without connection regeneration for ResNet-20 on CIFAR-10. The vertical red lines refer to the points when the learning rate is decayed. The pruning method is uniform pruning.

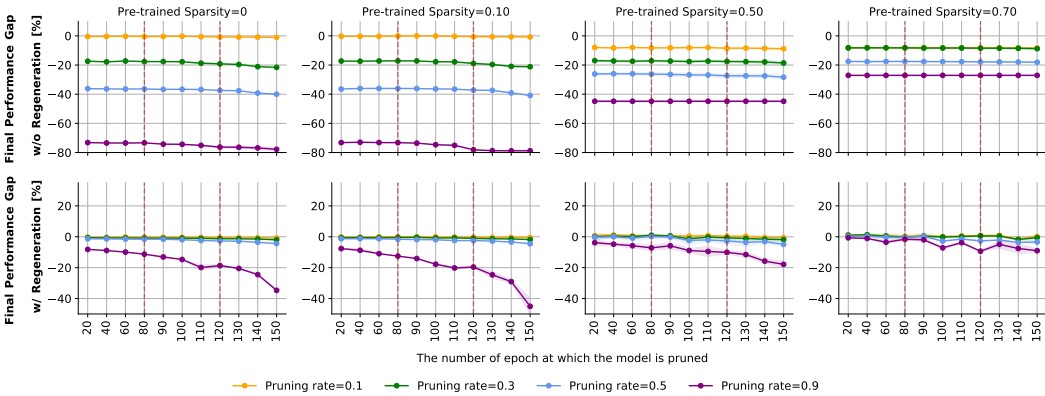

Figure 6: **Structured Pruning:** Final performance gap between the unpruned models and the pruned models for ResNet-20 on CIFAR-10. The vertical red lines refer to the points when the learning rate is decayed.

## A.2  VGG-19 on CIFAR-10 with Structured Filter Pruning

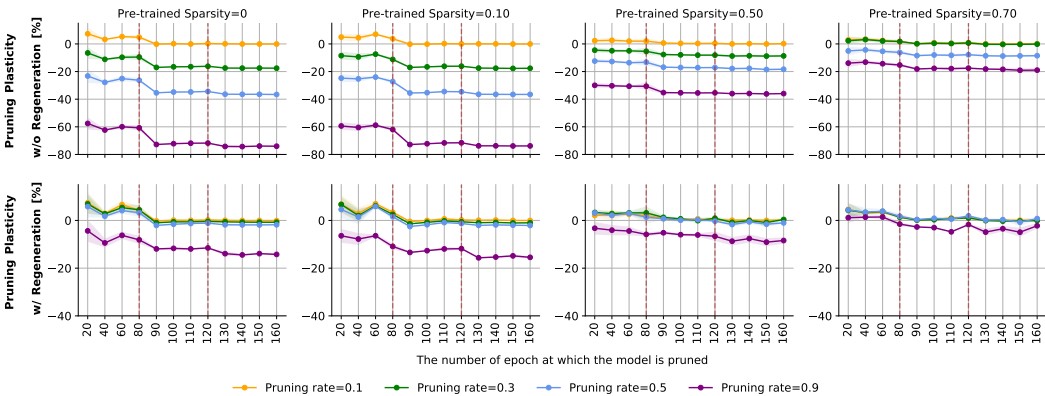

Figure 7: **Structured Pruning:** Pruning plasticity of under a 30-epochs continued training with and without connection regeneration for VGG-19 on CIFAR-10. The vertical red lines refer to the points when the learning rate is decayed. The pruning method is uniform pruning.

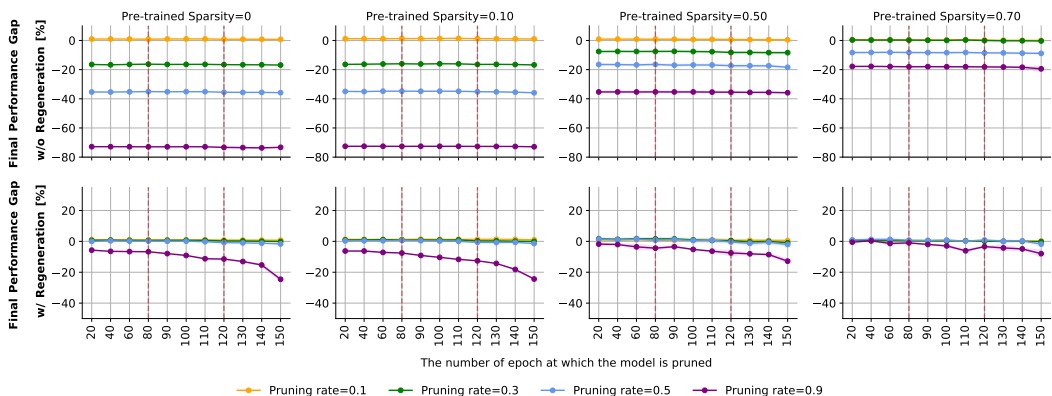

Figure 8: **Structured Pruning:** Final performance gap between the unpruned models and the pruned models for VGG-19 on CIFAR-10. The vertical red lines refer to the points when the learning rate is decayed.

## A.3   ResNet-20 on CIFAR-10 with Unstructured Uniform Pruning

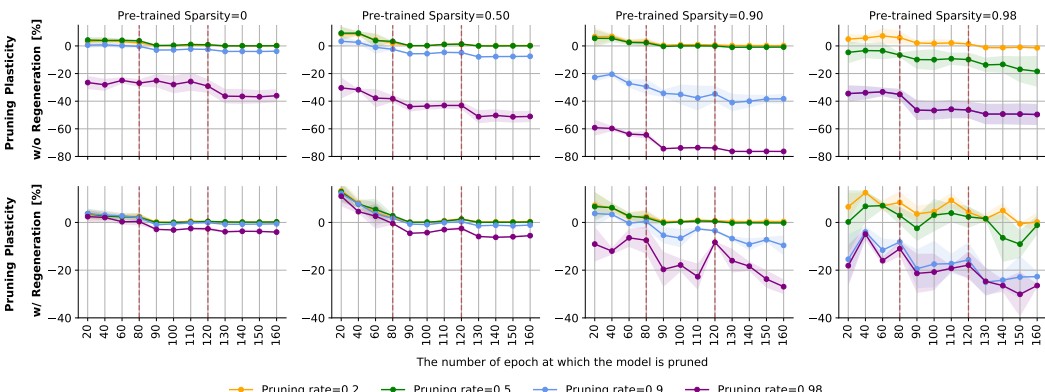

Figure 9: **Unstructured Pruning:** Pruning plasticity under a 30-epochs continued training with and without connection regeneration for ResNet-20 on CIFAR-10. The vertical red lines refer to the points when the learning rate is decayed.

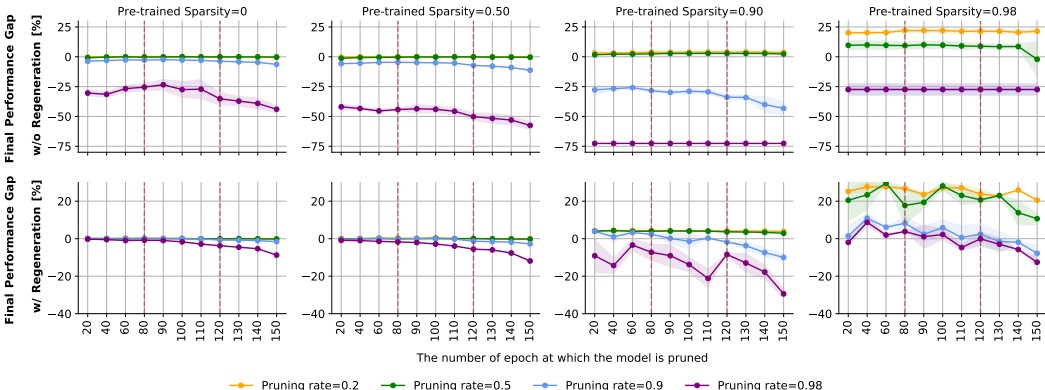

Figure 10: **Unstructured Pruning:** Final performance gap between the unpruned models and the pruned models for ResNet-20 on CIFAR-10. The vertical red lines refer to the points when the learning rate is decayed. The pruning method is uniform pruning.

## A.4 VGG-19 on CIFAR-10 with Unstructured Global Pruning

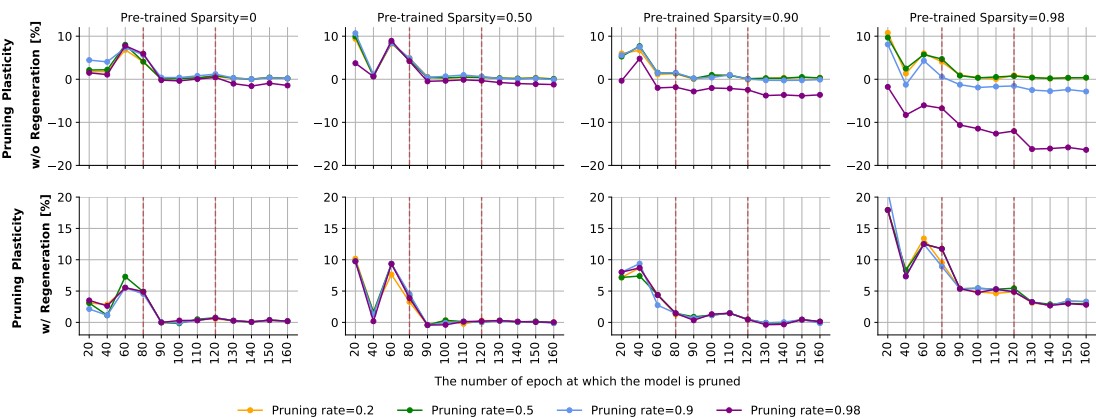

Figure 11: **Unstructured Pruning:** Pruning plasticity under a 30-epochs continued training with and without connection regeneration for VGG-19 on CIFAR-10. The vertical red lines refer to the points when the learning rate is decayed. The pruning method is global pruning.

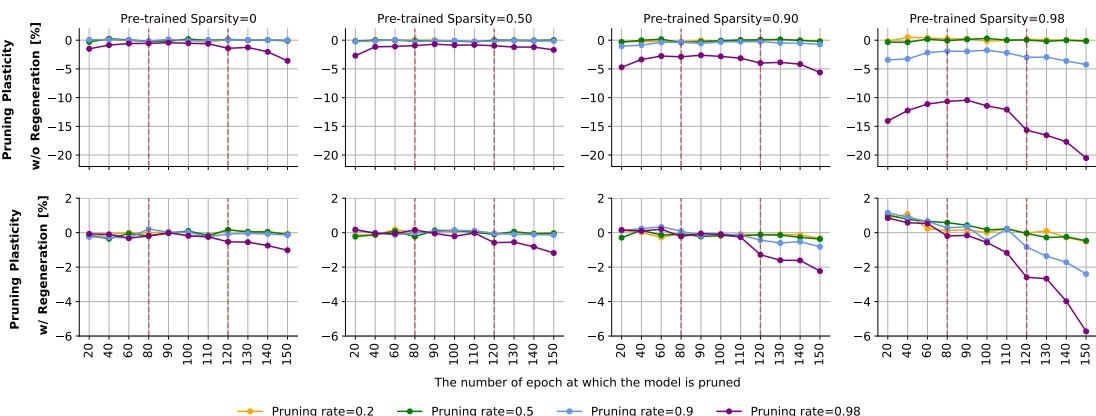

Figure 12: **Unstructured Pruning:** Final performance gap between the unpruned models and the pruned models for VGG-19 on CIFAR-10. The vertical red lines refer to the points when the learning rate is decayed. The pruning method is global pruning.

## A.5 VGG-19 on CIFAR-10 with Unstructured Uniform Pruning

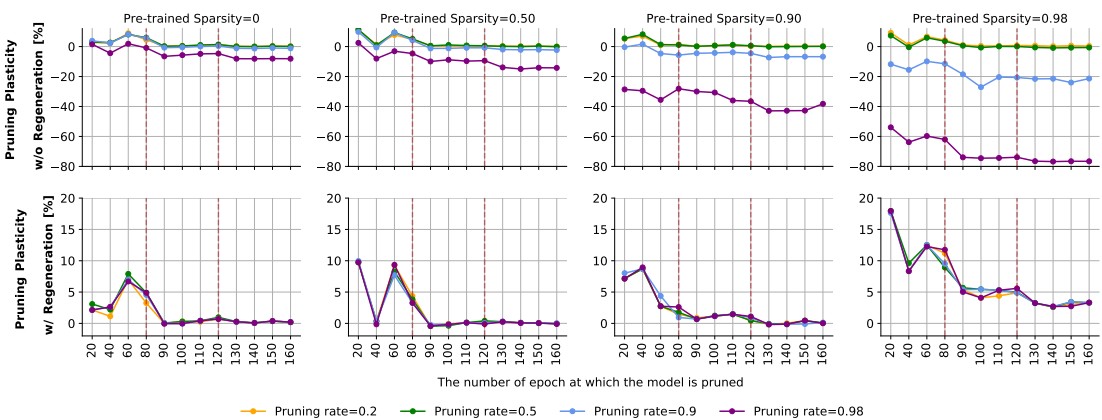

Figure 13: **Unstructured Pruning:** Pruning plasticity under a 30-epochs continued training with and without connection regeneration for VGG-19 on CIFAR-10. The vertical red lines refer to the points when the learning rate is decayed. The pruning method is uniform pruning.

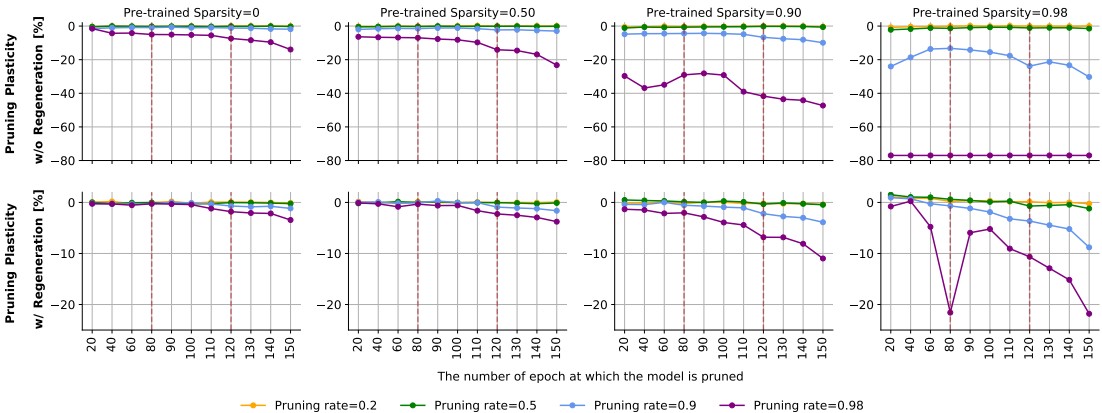

Figure 14: **Unstructured Pruning:** Final performance gap between the unpruned models and the pruned models for VGG-19 on CIFAR-10. The vertical red lines refer to the points when the learning rate is decayed. The pruning method is uniform pruning.

# B Implementation Details of GraNet

In this appendix, we share in detail the pseudocode and implementation details of GraNet.

## B.1 Algorithm

The pseudocode of GraNet is shared in Algorithm 1. The only difference between sparse-to-sparse training and dense-to-sparse training is the choices of initial sparsity $s_i$. For dense-to-sparse training, we need to set the initial sparsity of the model $s_i = 0$. To perform sparse-to-sparse training, we need to make sure the model is sparse at the beginning by setting the initial sparsity larger than 0, i.e., $s_i > 0$.

---
**Algorithm 1** The pseudocode of GraNet.

---
**Require:** Model weights $W \in \mathbb{R}^d$, initial sparsity $s_i$, target sparsity $s_f$, gradual pruning starting point $t_0$, gradual pruning end point $t_f$, gradual pruning frequency $\Delta T$.
1: $W \leftarrow$ randomly initialize $W$ with initial sparsity $s_i$
2: **for** each training step $t$ **do**
3:     training $W \leftarrow \text{SGD}(W)$
4:     **if** $t_o \leq t \leq t_f$ **and** $(t \bmod \Delta \text{T}) == 0$ **then**
5:         gradual pruning with the pruning rate produced by Eq. 1
6:         zero-cost neuroregeneration with Eq. 2 and Eq. 3
7:     **end if**
8: **end for**

---

## B.2 Hyperparameters

We share the hyperparameter choices in our experiments in Table 7.

Table 7: Experiment hyperparameters of GraNet used in this paper. Learning Rate (LR), Batch Size (BS), Epochs, Learning Rate Drop (LR Drop), Weight Decay (WD), Sparse Initialization (Sparse Init), Gradual Pruning Frequency ($\Delta T$), Initial Sparsity ($s_i$), Starting Epoch of Gradual Pruning ($t_0$), End Epoch of Gradual Pruning ($t_f$), Initial Neuroregeneration Ratio ($r$), Neuroregeneration Ratio ($r$ Sche), etc.

| Model | Data | Methods | LR | BS | Epochs | LR Drop, Epochs | WD | Sparse Init | Gradual Pruning | | | | Neuroregeneration | |
| --- | --- | --- | --- | --- | --- | --- | --- | --- | --- | --- | --- | --- | --- | --- |
| | | | | | | | | | $\Delta T$ | $s_i$ | $t_0$ | $t_f$ | $r$ | $r$ Sche |
| VGG-19 | CIFAR-10/100 | dense-to-sparse | 0.1 | 128 | 160 | 10x, [80, 120] | 5e-4 | Dense | 1000 | 0 | 0 epoch | 110 epoch | 0.5 | Cosine |
| | CIFAR-10/100 | sparse-to-sparse | 0.1 | 128 | 160 | 10x, [80, 120] | 5e-4 | ERK | 1000 | 0.5 | 0 epoch | 80 epoch | 0.5 | Cosine |
| ResNet-50 | CIFAR-10/100 | dense-to-sparse | 0.1 | 128 | 160 | 10x, [80, 120] | 5e-4 | Dense | 1000 | 0 | 0 epoch | 110 epoch | 0.5 | Cosine |
| | CIFAR-10/100 | sparse-to-sparse | 0.1 | 128 | 160 | 10x, [80, 120] | 5e-4 | ERK | 1000 | 0.5 | 0 epoch | 80 epoch | 0.5 | Cosine |
| | ImageNet | dense-to-sparse | 0.1 | 64 | 100 | 10x, [30, 60, 90] | 1e-4 | Dense | 4000 | 0 | 0 epoch | 30 epoch | 0.5 | Cosine |
| | ImageNet | sparse-to-sparse | 0.1 | 64 | 100 | 10x, [30, 60, 90] | 1e-4 | ERK | 4000 | 0.5 | 0 epoch | 30 epoch | 0.5 | Cosine |

## B.3 Implementation

The implementation used in the paper is modified based on the open-source code of Sparse Momentum repository[2] introduced by [8]. We added VGG-19 with batchnorm from the GraSP repository[3]. The code for calculating the inference FLOPs of ResNet-50 on ImageNet is modified based on the open-source code provided in the rethinking-network-pruning repository[4]. For the training FLOPs, we follow the way of approximating the training FLOPs of RigL [9], where the FLOPs of the backward pass are around twice the ones of the forward pass.

---
[2]https://github.com/TimDettmers/sparse_learning
[3]https://github.com/alecwangcq/GraSP
[4]https://github.com/Eric-mingjie/rethinking-network-pruning/blob/master/imagenet/weight-level/compute_flops.py

# C  Implementation Details of GMP

In this appendix, we share in detail the pseudocode and implementation details of GMP.

## C.1  Algorithm

Gradual Magnitude Pruning (GMP), introduced in [77] and studied further in [13], gradually sparsifies the neural network during the training process until the desired sparsity is reached. The pruning rate of each pruning iteration is:

$$s_t = s_f + (s_i - s_f)\left(1 - \frac{t - t_0}{n\Delta t}\right)^3 \quad t \in \{t_0, t_0 + \Delta t, ..., t_0 + n\Delta t\} \tag{4}$$

The pseudocode of GMP is shown in Algorithm 2.

---

**Algorithm 2** The pseudocode of GMP.

---

**Require:** Model weights $W \in \mathbb{R}^d$, initial sparsity $s_i$, target sparsity $s_f$, gradual pruning starting point $t_0$, gradual pruning end point $t_f$, gradual pruning frequency $\Delta T$.
1: $W \leftarrow$ randomly initialize $W$ with initial sparsity $s_i$
2: **for** each training step $t$ **do**
3:     training $W \leftarrow \text{SGD}(W)$
4:     **if** $t_o \leq t \leq t_f$ **and** $(t \bmod \Delta\text{T}) == 0$ **then**
5:         gradual pruning with the pruning rate produced by Eq. 1
6:     **end if**
7: **end for**

---

## C.2  Hyperparameters

To demonstrate the effectiveness of Zero-Cost Neuroregeneration, we reproduce GMP with our implementation for CIFAR-10/100 so that the only difference between GMP and GraNet is the Zero-Cost Neuroregeneration. Hence, all the hyperparameters of GMP on CIFAR-10/100 are the same as GraNet.

It is surprising that the number of training FLOPs required by GraNet is smaller than GMP reported in [13]. Since the authors of [13] did not share the specific hyperparameters that used to produce the results of GMP, we guess the pruning of GMP happens late in training. Thus, it makes sense that GraNet requires fewer training FLOPs than GMP, as the dense training time of GraNet is much shorter than GMP.

## C.3  Implementation

We reproduce GMP only for the results of CIFAR-10/100 for a fair comparison with GraNet. The results of GMP with ResNet-50 on ImageNet are obtained directly from [13].

We also test our implemented GMP with ResNet-50 on ImageNet. However, the performance is much worse than the results in [13] as shown below.

Table 8: Test accuracy of GMP ResNet-50 on ImageNet dataset with our own implementation.

| Method | Top-1 Accuracy | FLOPs (Train) | FLOPs (Test) | TOP-1 Accuracy | FLOPs (Train) | FLOPs (Train) |
|---|---|---|---|---|---|---|
| Dense | 76.8±0.09 | 1x (3.2e18) | 1x (8.2e9) | | | |
| | | S = 0.8 | | | S = 0.9 | |
| GMP [13] | 75.6 | 0.56× | 0.23× | 73.9 | 0.51× | 0.10× |
| GMP (our implementation) | 74.6 | 0.34× | 0.28× | 73.3 | 0.23× | 0.16× |

We are aware that our GMP implementation has several differences from the original Tensorflow implementation used by [77, 13]. Firstly, since our implementation reset the weight values to zero once the weights are pruned, the pruned weights of GMP are also set to zero. However, in the original

GMP implementation, only the masks are set to zero and the weight values are kept, leading to a situation where the pruned weights can be regenerated back in a natural way. Secondly, the original GMP uses uniform pruning and keeps the first layer dense and the sparsity of the last layer no larger than 0.8. Same as GraNet, our implementation of GMP prune all the layers including the first layer and the last layer.

We also compare GMP and GraNet with uniform pruning as used in [13], as shown below. While the results with CIFAR-10 in unclear, GraNet outperforms GMP with CIFAR-100 consistently. As we expected, the performance using uniform pruning is generally worse than global pruning.

Table 9: Test accuracy of pruned VGG-19 and ResNet32 on CIFAR-10 and CIFAR-100 datasets using uniform pruning.

| Dataset | CIFAR-10 | | | CIFAR-100 | | |
|---|---|---|---|---|---|---|
| Pruning ratio | 90% | 95% | 98% | 90% | 95% | 98% |
| **VGG-19** (Dense) | 93.85±0.05 | - | - | 73.43±0.08 | - | - |
| GMP [13] | **93.28±0.04** | 92.76±0.10 | 91.64±0.18 | 71.88±0.29 | 71.02±0.27 | 66.16±0.23 |
| GraNet ($s_i = 0$) | 93.12±0.03 | **92.88±0.08** | **91.87±0.06** | **72.37±0.01** | **71.48±0.25** | **70.14±0.18** |
| **ResNet-50** (Dense) | 94.75±0.01 | - | - | 78.23±0.18 | - | - |
| GMP [13] | 94.08±0.14 | **94.20±0.24** | **93.66±0.44** | 77.30±0.27 | 76.77±0.02 | 75.38±0.24 |
| GraNet ($s_i = 0$) | **94.19±0.23** | 94.16±0.26 | 93.64±0.25 | **77.57±0.12** | **77.15±0.18** | **76.17±0.15** |

# D ResNet-50 Learnt Budgets and Backbone Sparsities

Table 10 summarizes the final sparsity budgets for 90% sparse ResNet50 on ImageNet-1K obtained by various methods. Backbone represents the sparsity budgets for all the CNN layers without the last fully-connected layer. VD refers to Variational Dropout [45] and GS refers to iterative magnitude pruning using a global threshold for global sparsity [18].

Table 10: ResNet-50 Learnt Budgets and Backbone Sparsities at Sparsity 0.9

| Metric | Fully Dense Params | Fully Dense FLOPs | Sparsity (%) | | | | | | | |
| --- | --- | --- | --- | --- | --- | --- | --- | --- | --- | --- |
| | | | GraNet ($s_i = 0$) | GraNet ($s_i = 0.5$) | STR | Uniform | ERK | SNFS | VD | GS |
| Overall | 25502912 | 8178569216 | 89.99 | 89.98 | 90.23 | 90.00 | 90.07 | 90.06 | 90.27 | 89.54 |
| Backbone | 23454912 | 8174272512 | 89.89 | 90.65 | 92.47 | 90.00 | 89.82 | 89.44 | 91.41 | 90.95 |
| Layer 1 - conv1 | 9408 | 118013952 | 53.50 | 40.60 | 59.80 | 90.00 | 58.00 | 2.50 | 31.39 | 35.11 |
| Layer 2 - layer1.0.conv1 | 4096 | 236027904 | 54.60 | 43.40 | 83.28 | 90.00 | 0.00 | 2.50 | 39.50 | 56.05 |
| Layer 3 - layer1.0.conv2 | 36864 | 231211008 | 78.80 | 64.50 | 89.48 | 90.00 | 82.00 | 2.50 | 67.87 | 75.04 |
| Layer 4 - layer1.0.conv3 | 16384 | 102760448 | 78.00 | 67.40 | 85.80 | 90.00 | 4.00 | 2.50 | 64.87 | 70.31 |
| Layer 5 - layer1.0.downsample.0 | 16384 | 102760448 | 79.30 | 72.90 | 83.34 | 90.00 | 4.00 | 2.50 | 60.38 | 66.88 |
| Layer 6 - layer1.1.conv1 | 16384 | 102760448 | 76.60 | 67.30 | 89.89 | 90.00 | 4.00 | 2.50 | 61.35 | 75.09 |
| Layer 7 - layer1.1.conv2 | 36864 | 231211008 | 76.70 | 62.10 | 90.60 | 90.00 | 82.00 | 2.50 | 64.38 | 80.42 |
| Layer 8 - layer1.1.conv3 | 16384 | 102760448 | 74.10 | 54.50 | 91.70 | 90.00 | 4.00 | 2.50 | 65.83 | 80.00 |
| Layer 9 - layer1.2.conv1 | 16384 | 102760448 | 72.20 | 58.80 | 88.07 | 90.00 | 4.00 | 2.50 | 68.75 | 75.21 |
| Layer 10 - layer1.2.conv2 | 36864 | 231211008 | 72.70 | 58.50 | 87.03 | 90.00 | 82.00 | 2.50 | 70.86 | 74.95 |
| Layer 11 - layer1.2.conv3 | 16384 | 102760448 | 73.20 | 57.30 | 90.99 | 90.00 | 4.00 | 2.50 | 54.05 | 79.28 |
| Layer 12 - layer2.0.conv1 | 32768 | 205520896 | 68.30 | 49.30 | 85.95 | 90.00 | 43.00 | 2.50 | 57.10 | 70.89 |
| Layer 13 - layer2.0.conv2 | 147456 | 231211008 | 77.50 | 69.10 | 93.91 | 90.00 | 91.00 | 62.90 | 78.65 | 85.39 |
| Layer 14 - layer2.0.conv3 | 65536 | 102760448 | 71.70 | 61.10 | 93.13 | 90.00 | 52.00 | 11.00 | 85.49 | 83.54 |
| Layer 15 - layer2.0.downsample.0 | 131072 | 205520896 | 90.30 | 86.80 | 94.96 | 90.00 | 71.00 | 66.10 | 79.96 | 88.36 |
| Layer 16 - layer2.1.conv1 | 65536 | 102760448 | 85.20 | 83.00 | 95.31 | 90.00 | 52.00 | 32.60 | 72.07 | 88.25 |
| Layer 17 - layer2.1.conv2 | 147456 | 231211008 | 85.30 | 81.10 | 91.50 | 90.00 | 91.00 | 61.60 | 84.41 | 85.37 |
| Layer 18 - layer2.1.conv3 | 65536 | 102760448 | 80.00 | 68.60 | 93.66 | 90.00 | 52.00 | 20.80 | 79.19 | 86.53 |
| Layer 19 - layer2.2.conv1 | 65536 | 102760448 | 82.60 | 80.70 | 94.61 | 90.00 | 52.00 | 29.10 | 73.94 | 86.40 |
| Layer 20 - layer2.2.conv2 | 147456 | 231211008 | 83.20 | 82.40 | 94.86 | 90.00 | 91.00 | 63.90 | 78.48 | 88.29 |
| Layer 21 - layer2.2.conv3 | 65536 | 102760448 | 79.30 | 76.40 | 93.38 | 90.00 | 52.00 | 22.90 | 78.09 | 85.87 |
| Layer 22 - layer2.3.conv1 | 65536 | 102760448 | 81.10 | 77.10 | 93.26 | 90.00 | 52.00 | 27.60 | 78.66 | 84.87 |
| Layer 23 - layer2.3.conv2 | 147456 | 231211008 | 82.10 | 83.40 | 93.21 | 90.00 | 91.00 | 65.30 | 84.38 | 87.14 |
| Layer 24 - layer2.3.conv3 | 65536 | 102760448 | 82.40 | 77.30 | 94.14 | 90.00 | 52.00 | 25.70 | 82.07 | 86.84 |
| Layer 25 - layer3.0.conv1 | 131072 | 205520896 | 72.80 | 61.00 | 88.85 | 90.00 | 71.00 | 48.70 | 66.56 | 78.40 |
| Layer 26 - layer3.0.conv2 | 589824 | 231211008 | 84.60 | 83.30 | 96.14 | 90.00 | 96.00 | 90.20 | 87.92 | 92.93 |
| Layer 27 - layer3.0.conv3 | 262144 | 102760448 | 78.00 | 69.70 | 93.19 | 90.00 | 76.00 | 73.30 | 92.19 | 86.19 |
| Layer 28 - layer3.0.downsample.0 | 524288 | 205520896 | 95.30 | 95.20 | 97.20 | 90.00 | 86.00 | 93.70 | 88.76 | 94.66 |
| Layer 29 - layer3.1.conv1 | 262144 | 102760448 | 91.30 | 91.40 | 95.36 | 90.00 | 76.00 | 81.10 | 91.79 | 93.60 |
| Layer 30 - layer3.1.conv2 | 589824 | 231211008 | 91.10 | 93.10 | 95.06 | 90.00 | 96.00 | 90.40 | 92.47 | 93.07 |
| Layer 31 - layer3.1.conv3 | 262144 | 102760448 | 85.10 | 81.50 | 94.84 | 90.00 | 76.00 | 78.10 | 88.88 | 90.54 |
| Layer 32 - layer3.2.conv1 | 262144 | 102760448 | 90.10 | 89.70 | 96.77 | 90.00 | 76.00 | 80.40 | 84.86 | 93.44 |
| Layer 33 - layer3.2.conv2 | 589824 | 231211008 | 90.10 | 93.40 | 95.59 | 90.00 | 96.00 | 90.80 | 91.50 | 93.73 |
| Layer 34 - layer3.2.conv3 | 262144 | 102760448 | 86.70 | 83.90 | 94.99 | 90.00 | 76.00 | 79.30 | 81.59 | 91.13 |
| Layer 35 - layer3.3.conv1 | 262144 | 102760448 | 89.20 | 91.00 | 96.08 | 90.00 | 76.00 | 80.70 | 76.64 | 93.18 |
| Layer 36 - layer3.3.conv2 | 589824 | 231211008 | 90.90 | 94.20 | 96.10 | 90.00 | 96.00 | 90.70 | 91.26 | 93.63 |
| Layer 37 - layer3.3.conv3 | 262144 | 102760448 | 88.50 | 87.50 | 94.94 | 90.00 | 76.00 | 79.00 | 85.46 | 91.63 |
| Layer 38 - layer3.4.conv1 | 262144 | 102760448 | 88.90 | 89.60 | 95.49 | 90.00 | 76.00 | 79.40 | 85.33 | 91.98 |
| Layer 39 - layer3.4.conv2 | 589824 | 231211008 | 92.20 | 94.70 | 95.66 | 90.00 | 96.00 | 91.00 | 91.57 | 94.21 |
| Layer 40 - layer3.4.conv3 | 262144 | 102760448 | 90.30 | 88.60 | 94.49 | 90.00 | 76.00 | 79.00 | 86.19 | 91.63 |
| Layer 41 - layer3.5.conv1 | 262144 | 102760448 | 88.30 | 87.50 | 95.09 | 90.00 | 76.00 | 78.30 | 84.64 | 90.72 |
| Layer 42 - layer3.5.conv2 | 589824 | 231211008 | 92.30 | 94.90 | 94.92 | 90.00 | 96.00 | 91.00 | 91.14 | 93.43 |
| Layer 43 - layer3.5.conv3 | 262144 | 102760448 | 89.20 | 87.90 | 93.14 | 90.00 | 76.00 | 78.20 | 84.09 | 89.56 |
| Layer 44 - layer4.0.conv1 | 524288 | 205520896 | 80.20 | 72.80 | 90.32 | 90.00 | 86.00 | 85.80 | 77.90 | 85.35 |
| Layer 45 - layer4.0.conv2 | 2359296 | 231211008 | 89.80 | 93.60 | 95.66 | 90.00 | 98.00 | 97.60 | 96.53 | 95.07 |
| Layer 46 - layer4.0.conv3 | 1048576 | 51380224 | 84.70 | 82.40 | 91.14 | 90.00 | 88.00 | 93.20 | 93.52 | 89.21 |
| Layer 47 - layer4.0.downsample.0 | 2097152 | 205520896 | 99.00 | 99.20 | 96.79 | 90.00 | 93.00 | 98.80 | 93.80 | 96.72 |
| Layer 48 - layer4.1.conv1 | 1048576 | 102760448 | 93.10 | 95.60 | 93.69 | 90.00 | 88.00 | 94.10 | 94.96 | 92.69 |
| Layer 49 - layer4.1.conv2 | 2359296 | 231211008 | 93.60 | 97.30 | 93.98 | 90.00 | 98.00 | 97.70 | 97.76 | 93.85 |
| Layer 50 - layer4.1.conv3 | 1048576 | 102760448 | 90.30 | 90.80 | 90.48 | 90.00 | 88.00 | 94.20 | 94.53 | 89.84 |
| Layer 51 - layer4.2.conv1 | 1048576 | 205520896 | 87.30 | 87.10 | 87.57 | 90.00 | 88.00 | 93.60 | 94.19 | 85.91 |
| Layer 52 - layer4.2.conv2 | 2359296 | 231211008 | 91.70 | 96.80 | 84.37 | 90.00 | 98.00 | 97.90 | 94.92 | 87.14 |
| Layer 53 - layer4.2.conv3 | 1048576 | 102760448 | 85.00 | 83.40 | 80.29 | 90.00 | 88.00 | 94.50 | 89.64 | 80.65 |
| Layer 54 - fc | 2048000 | 4096000 | 91.30 | 82.40 | 64.50 | 90.00 | 93.00 | 97.10 | 77.17 | 73.43 |

# E  FLOPs Dynamics During Training with ResNet-50 on ImageNet

To have an overview of how the FLOPs of the pruned model evolves during training, we share the FLOPs dynamics (inference on single sample) of the pruned ResNet-50 during the course of training in Figure 15. While starting from a model with a higher number of FLOPs compared with GraNet ($s_i = 0.5$), GraNet ($s_i = 0$) is gradually sparsified towards a sparse structure with lower FLOPs.

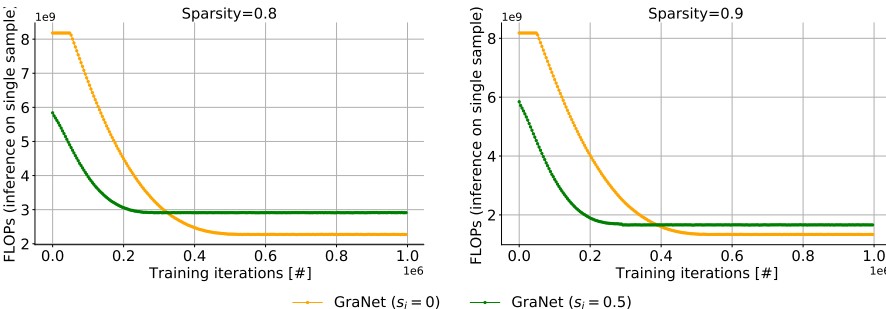

Figure 15: FLOPs dynamics (inference on single sample) of GraNet with ResNet-50 on ImageNet during training.