# OpenReview forum: "Sparse Training via Boosting Pruning Plasticity  with Neuroregeneration"
_NeurIPS.cc/2021/Conference — NeurIPS 2021 Poster_

### Official Review · Reviewer_93rT · 2021-07-13

**Rating:** 7
**Confidence:** 5

**Summary:**

This work aims to combine ideas presented in recent dynamic sparse training (DST) methods with gradual magnitude pruning. Additionally the paper has some novel experiments that confirm previous work on when-is-best to prune during training. Although an interesting and potentially impactful idea, the story and results need a bit more work.

**Limitations And Societal Impact:**

Yes.

**Main Review:**

# Pros
1) Results in Fig:2 are quite interesting. Though some clarification is needed (see below)
2) The idea of combining pruning and re-generation is novel and authors do a good job on comparing with a large set of methods.

# Cons
1) "For this reason, we determine to focus on gradient-based regeneration, i.e., regenerating the same number of connections as pruned with the largest gradient magnitude." This is sounds like what [rigl] does for training sparse networks. Is the method proposed for regeneration is practically same as rigl? If so it would be nice to called it so. If not, it would be nice to see a discussion on similarities and differences. My understanding is GraNet=GMP+RigL. It would be nice to make this clear.
2) It needs to be clear which experiments uses global pruning (and thus non-uniform sparsity). For example on Table-3, GraNet-ST (ERK) requires 0.37× training flops which is less than rigl (erk). Shouldn't be the training flops of GraNet-ST always larger than rigl since it starts with more parameters? I am assuming global pruning is not used here, as if so; then the resulting distribution is not ERK anymore and this should be made clear.
3) It should be make clear that GraNet-ST is different than other sparse training methods like set and rigl in the sense that it requires more sparsity/memory to begin with.
3) I couldn't see any ablations on the initial sparsity of GraNet-ST method. I think the value 0.5 is picked, which is a significant increase in parameter count if the final sparsity is >90%; it would be nice to see effect of initial sparsity.
4) It might be better to use term 'sparsity' instead of 'density' on Figure 2. More importantly I am confused what "pruning rate" is referring to. Reading the text, it seems to refer sparsity of the network after pruning. Then it is not clear how do you prune a 98% sparse (2% dense) network to 50% sparsity. Do you only grow? This needs to be cleared I think. Experiments in Figure:2 looks quite interesting otherwise.

## Minor
1) "To better understand the effect of pruning during the optimization proces" [zhu2017] is a relevant work that studies the same.
2) "To the best of our knowledge, this is the first time in the literature that sparse-to-sparse training reaches a test accuracy of 76% with ResNet-50 on ImageNet at sparsity 0.8." [rigl] paper I think has 5x training runs with such results. If extended training runs are not considered; this should be explained. I think a better way to compare dense-to-sparse methods with DST methods is to use training flops on the x-axis.
3) "Perhaps most impressively, the latter for the first time boosts the sparse-to-sparse training performance over various dense-to-sparse methods by a large margin with ResNet-50 on ImageNet." I guess large-margin is subjective but [rigl] shows the same.

[rigl] https://arxiv.org/abs/1911.11134
[zhu2017] https://arxiv.org/pdf/1710.01878.pdf

## After Rebuttal
I thank the authors for their strong rebuttal. I think all my concerns/questions are addressed and I believe this work is a timely work that inspire and impact future research. Therefore I support acceptance.

One final suggestion:
- I think it might help the readers to name both versions of the method as GraNet as when initial sparsity=0 GraNet-ST is GraNet. Then authors can use sth like GraNet (s=0) to identify pruning version.


**Time Spent Reviewing:**

6

---

> ### Author Response · Authors · 2021-08-10
> **Response to Reviewer 93rT  (2/2)**
>
> ## Minor questions:
>
> &nbsp;
>
> ### Q7: "To better understand the effect of pruning during the optimization proces" [zhu2017] is a relevant work that studies the same.
>
> * Zhu2017 is a great related work that we have cited in our submission. We are glad to include a paragraph of discussion about this paper in the next version.
>
>  ### Q8: "To the best of our knowledge, this is the first time in the literature that sparse-to-sparse training reaches a test accuracy of 76% with ResNet-50 on ImageNet at sparsity 0.8." [rigl] paper I think has 5x training runs with such results. If extended training runs are not considered; this should be explained. I think a better way to compare dense-to-sparse methods with DST methods is to use training flops on the x-axis.
>
> * For this claim, we mean training a sparse neural network from scratch for a typical training time, i.e., ~ 100 epochs without extended training runs, which is widely used in recent sparse training literature [1, 2, 5, 6, 7]. We will make this clearer.
>
>  ### Q9: "Perhaps most impressively, the latter for the first time boosts the sparse-to-sparse training performance over various dense-to-sparse methods by a large margin with ResNet-50 on ImageNet." I guess large-margin is subjective but [rigl] shows the same.
>
> * We fully agree that RigL is the first work that demonstrates that the sparse-to-sparse training method can outperform the current best dense-to-sparse training method. We will re-emphasize the contribution of RigL in our next version and adjust our claim as something like “We are the first work to outperform various dense-to-sparse training methods without extending the training time”.
>
> * Meanwhile, it is perhaps fair to say “large-margin” for our method, as GraNet-ST improves the sparse training performance over RigL by a  >1% margin with ResNet50 on ImageNet at an extreme sparsity of 90%.
>
> &nbsp;
>
> [1]  [Evci, Utku, et al. "Rigging the lottery: Making all tickets winners."](https://arxiv.org/abs/1911.11134)
>
> [2]  [Liu, Shiwei, et al. "Do we actually need dense over-parameterization? in-time over-parameterization in sparse training."](https://arxiv.org/abs/2102.02887)
>
> [3]  [Frankle, Jonathan, David J. Schwab, and Ari S. Morcos. "The early phase of neural network training."](https://arxiv.org/abs/2002.10365)
>
> [4]  [Frankle, Jonathan, et al. "Linear mode connectivity and the lottery ticket hypothesis."](https://arxiv.org/abs/1912.05671)
>
> [5]  [Dettmers, Tim, and Luke Zettlemoyer. "Sparse networks from scratch: Faster training without losing performance."](https://arxiv.org/abs/1907.04840)
>
> [6]  [Mostafa, Hesham, and Xin Wang. "Parameter efficient training of deep convolutional neural networks by dynamic sparse reparameterization." ](https://arxiv.org/abs/1902.05967)
>
> [7]  [Jayakumar, Siddhant M., et al. "Top-kast: Top-k always sparse training."  ](https://arxiv.org/abs/2106.03517)

---

> ### Author Response · Authors · 2021-08-10
> **Response to Reviewer 93rT  (1/2)**
>
> We thank the reviewer for recognizing that our findings are interesting and potentially impactful. Your constructive comments will make our paper clearer. Below we clarify your questions and we sincerely hope they could address all your concerns:
>
> ### Q1: "For this reason, we determine to focus on gradient-based regeneration, i.e., regenerating the same number of connections as pruned with the largest gradient magnitude." This sounds like what [rigl] does for training sparse networks. Is the method proposed for regeneration is practically same as rigl? My understanding is GraNet=GMP+RigL. It would be nice to make this clear.
>
> * Yes, the neuroregeneration method of our paper is the same as the growth criterion used in RigL. We were definitely going to cite RigL when introducing the gradient-based regeneration, just like citing SET when introducing random-based regeneration, but missed to do so due to our negligence. We apologize for this and will make this very clear in the next version!
>
> * GraNet-ST is in general a combination of GMP and RigL, though with some important differences (please refer to the next question for the difference details). The experimental results show that the proposed combination of GMP and RigL is greater than the sum of its parts.
>
> * GMP allows RigL to start from a denser network, and thus to explore more parameters in the early training phase. This denser initial network is crucial for the improved performance since the In-Time Over-Parameterization [1] paper has pointed out the strong correlation between the total number of visited parameters and the test accuracy. The early training phase is also very important for sparse networks to achieve matching performance [2, 3]. The experiments for the initial sparsity question below have clearly shown this.
>
> * Based on the insights given by pruning plasticity, we use RigL to further optimize the sparse networks discovered by GMP in both dense-to-sparse and sparse-to-sparse settings. The better mask positions found by our method over GMP confirm the values of RigL in the dense-to-sparse training scenario too. We believe this does present novel insights into the behavior of the existing algorithms.
>
> ### Q2: It needs to be clear which experiments uses global pruning (and thus non-uniform sparsity). For example on Table-3, GraNet-ST (ERK) requires 0.37× training flops which is less than rigl (erk). Shouldn't be the training flops of GraNet-ST always larger than rigl since it starts with more parameters? I am assuming global pruning is not used here, as if so; then the resulting distribution is not ERK anymore and this should be made clear.
>
> * Thank you for pointing this question out. This helps us to improve the clarity of our paper. We indeed use global pruning for GMP, so that we can find a better final distribution with fewer FLOPs and higher test accuracy over ERK. The “ERK” in the bracket of GraNet-ST is used to show that we initialize GraNet-ST with ERK distribution, which can be quite misleading and unexpectedly causes a misunderstanding. We will remove it in the next version.
>
> * The fewer training FLOPs of GraNet-ST come from two important technical differences: (1) better final sparse distribution discovered by magnitude global pruning; (2) short time of gradual pruning (the first 30 epochs for ResNet-50 on ImageNet). Although starting with more parameters, the global pruning helps GraNet-ST quickly evolve to a better sparsity distribution with lower feedforward FLOPs (S=0.8: GraNet-ST: 0.35× vs ERK: 0.42×). After the gradual pruning process ends, the network is trained with this better distribution for 70 epochs, so that the overhead in the early training phase with large training FLOPs is amortized by the later and longer training phase with fewer training FLOPs.
>
> ### Q3: It should be make clear that GraNet-ST is different than other sparse training methods like set and rigl in the sense that it requires more sparsity/memory to begin with.
>
> * We think this is a fair point, and we actually have mentioned this in the last paragraph of Section 4.2, page 7 as “Different from the existing DST methods in which the sparsity of the forward pass is fixed, GraNet-ST starts from a denser yet still sparse model and gradually prunes the sparse model to the desired sparsity.” We will make this clearer in the next version.
>
> ### Q4: I couldn't see any ablations on the initial sparsity of GraNet-ST method.  It would be nice to see the effect of initial sparsity.
>
> * Thank you for your valuable suggestion. This indeed helps us improve our method. As you suggested, we added experiments to study the effect of the initial sparsity. The initial sparsity is chosen from [0.3, 0.5, 0.6, 0.7, 0.8] and the final sparsity is fixed as 0.9. We can see the training FLOPs of GraNet-ST is quite robust to the initial sparsity. Surprisingly yet reasonably, it seems that the larger the initial model is, the better final sparsity distribution GraNet-ST can find, with higher test accuracy and fewer feedforward FLOPs. The lower feedforward FLOPs perfectly balance the overhead caused by the larger initial model.
>
> |  Methods | Model/Data | Initial Sparsity | Final Sparsity |  Test Acc [%]   |  Training FLOPs  |  Test FLOPs  |
> | ------------- | :-----------: |:-----------:|:-------------:|:-------------:|:-------------:|:-------------:|
> | GraNet-ST |  ResNet-50/ImageNet  |  30% | 90% | 74.6 | 0.24$\times$   | 0.18$\times$ |
> | GraNet-ST |  ResNet-50/ImageNet  |  50% | 90% | 74.5 | 0.25$\times$   | 0.20$\times$ |
> | GraNet-ST|  ResNet-50/ImageNet   |  60% | 90% | 74.4 | 0.25$\times$   | 0.22$\times$ |
> | GraNet-ST |  ResNet-50/ImageNet  |  70% | 90% | 74.2 | 0.24$\times$   | 0.22$\times$ |
> | GraNet-ST|  ResNet-50/ImageNet   |  80% | 90% | 74.1 | 0.25$\times$   | 0.24$\times$ |
> | RigL |  ResNet-50/ImageNet |  90% |  90% | 73.0| 0.25$\times$   | 0.24$\times$ |
>
>
>  ### Q5: I think the value 0.5 is picked, which is a significant increase in parameter count if the final sparsity is >90%;
>
> * We fully agree with your comment and we have conducted experiments with ResNet-50 on ImageNet to confirm this. The initial sparsity is set as **50%**. When the final sparsity is moderate (e.g., 80%, 90%), GraNet-ST requires a lower (or the same) number of training FLOPs than RigL, whereas GraNet-ST may require more training FLOPs than RigL when the final sparsity is high (95%, 96.5%). This makes sense since when the sparsity is very high, the saved FLOPs count of the better distribution discovered by GraNet is too small to amortize the overhead caused by large initial models. Yet, the increased number of training FLOPs of GraNet-ST leads to substantial accuracy improvement (> 2%) over RigL at extremely high sparsities.
>
> |  Methods | Model/Data | Initial Sparsity | Final Sparsity |  Test Acc [%]   |  Training FLOPs  |  Test FLOPs  |
> | ------------- | :-----------: |:-----------:|:-------------:|:-------------:|:-------------:|:-------------:|
> | RigL |  ResNet-50/ImageNet  |  50% | 80% |75.1 | 0.42$\times$   | 0.42$\times$ |
> | GraNet-ST |  ResNet-50/ImageNet  |  50% | 80%  |**76.0** | 0.37$\times$   | 0.35$\times$ |
> | RigL |  ResNet-50/ImageNet  |  50% | 90% | 73.0 | 0.25$\times$   | 0.24$\times$ |
> | GraNet-ST |  ResNet-50/ImageNet  |  50% | 90% |**74.5** | 0.25$\times$   | 0.20$\times$ |
> | RigL |  ResNet-50/ImageNet  |  50% |  95% | 69.7 | 0.12$\times$   | 0.12$\times$ |
> | GraNet-ST |  ResNet-50/ImageNet  |  50% | 95% |**72.3** | 0.17$\times$   | 0.12$\times$ |
> | RigL |  ResNet-50/ImageNet  |  50% | 96.5% | 67.2 | 0.11$\times$   | 0.11$\times$ |
> | GraNet-ST |  ResNet-50/ImageNet  |  50% | 96.5% |**70.5** | 0.15$\times$   | 0.09$\times$ |
>
>  ### Q6: It might be better to use term 'sparsity' instead of 'density' on Figure 2. More importantly, I am confused what "pruning rate" is referring to. Reading the text, it seems to refer sparsity of the network after pruning. Then it is not clear how do you prune a 98% sparse (2% dense) network to 50% sparsity. Do you only grow? This needs to be cleared I think. Experiments in Figure:2 looks quite interesting otherwise.
>
> * We will use sparsity in our new version. Here, we just keep it as density for better explanation. We study the pruning plasticity of both the sparse trained models and dense trained models. So that we need to pre-train models from scratch at various density levels, including 100%, 50%, 10%, and 2%, and measure their pruning plasticity. In Figure 2, the “Model Density = 100%”, …, “Model Density = 2%” refers to the density level (1 - sparsity) of these pre-trained models.
>
> * To measure the plasticity of these pre-trained models, we choose four different pruning rates noted as  “Pruning rate” in Figure 2, referring to how many percentages of weights are further pruned from these saved models for pruning plasticity. For example, in the last column of Figure 2, the green line (Pruning rate = 0.5) means that we further prune 50% weights from a 2% density ResNet20. Therefore, the remaining number of parameters of the final model is 2% × 50% = 1%.

---

> ### Author Response · Authors · 2021-08-22
> **Response to Reviewer 93rT**
>
> We really appreciate your valuable comments and your acceptance!
>
> All your comments are constructive, helping to make our paper stronger and clearer. We will name both versions of the method as GraNet and incorporate all the discussions in the next version.

---

### Official Review · Reviewer_RYno · 2021-07-17

**Rating:** 8
**Confidence:** 4

**Summary:**

This paper proposes a concept of pruning plasticity to study the effect of pruning during training. Using the insights obtained from the pruning plasticity, the authors improve the state-of-the-art sparse training performance. Since a large body of recent works has emerged to accelerate the training process, this work is helpful for this area.

The findings are interesting and make good connections with separately reported phenomena in the literature. Based on the insights from pruning plasticity, the authors further propose two gradual pruning methods which improve the performance over the existing methods.


**Limitations And Societal Impact:**

Yes

**Main Review:**

Originality:
(1)	The paper does a thorough study of pruning during training, which is indeed recently ignored by researchers compared with after-training pruning and before-training pruning. The recently emerged before-training pruning (pruning at initialization) absolutely receive a lot of attention. This work, however, doesn’t follow the crowd and highlights the values of an existing technique - pruning during training. Due to the failure of the existing before-training pruning methods to capture the important masks pointed by [1], discovering either early-existing matching subnetworks or more accurate subnetworks during training have a great value for the community.
(2)	The observations associated with pruning plasticity make good connections with several existing phenomena in the pruning literature, providing valuable insights to the people who want to use these techniques in reality.
(3)	With the insights provided by pruning plasticity and the neuroregeneration from the nervous system, authors modify the existing gradual magnitude pruning methods and demonstrate the state of the art sparse training performance as well as the gradual pruning performance.

Quality:
(1)	The paper is clearly written and easy to follow. The authors provide an adequate related work review.
(2)	I like the random initialization analysis of the discovered masks between GMP and GraNet, making the overall paper more robust.
(3)	The claim is well supported by the extensive experiments.

Significance:
(1)	While the idea of during-training pruning is not new, it indeed achieves the best performance-efficiency tradeoff compared with after-training pruning and before-training pruning. Works like this paper that provides better understanding and performance improvement of pruning during training are important for the ML community, providing good motivations for future hardware design.
(2)	The proposed methods are sound and performant. Especially, the 0.9% and 1.5% accuracy improvement achieved by GraNet-ST over RigL with ResNet50 on ImageNet at sparsity of 80% and 90%, respectively, is quite impressive. It is the first study which, up to my knowledge, shows consistently that dynamic sparse training can outperform dense-to-sparse training with even fewer training FLOPs.

Improvements:
(1)	This paper only considers unstructured pruning. I suppose that the findings of pruning plasticity can also generalize to the structured setting. It would be nice to test pruning plasticity on the structured pruning.
(2)	The authors should cite [2,3,4] for the zero-cost neuroregeneration.
(3)	It is unclear to me if the pruned weights are still updated or not during the backward pass. This would cause a significant computational difference. It would be nice to make this point clearer.
(4)	I suggest the authors to put more emphasis on the dynamic sparse training part (GraNet-ST) rather than GraNet, which is more promising in terms of the accuracy-efficiency trade-off. The description of GraNet-ST in the current version is somehow unclear, especially for readers who are not familiar with sparse training.

Reference:
[1] Frankle, Jonathan, et al. "Pruning neural networks at initialization: Why are we missing the mark?." arXiv preprint arXiv:2009.08576 (2020).
[2] Mocanu, Decebal Constantin, et al. "Scalable training of artificial neural networks with adaptive sparse connectivity inspired by network science." Nature communications 9.1 (2018): 1-12.
[3] Evci, Utku, et al. "Rigging the lottery: Making all tickets winners." International Conference on Machine Learning. PMLR, 2020.
[4] Dettmers, Tim, and Luke Zettlemoyer. "Sparse networks from scratch: Faster training without losing performance." arXiv preprint arXiv:1907.04840 (2019).


**Time Spent Reviewing:**

6 hours

---

> ### Author Response · Authors · 2021-08-10
> **Response to Reviewer RYno:**
>
> We are grateful that the reviewer likes our purpose and the potential significance of our work. We have responded to the reviewer’s comments in-line below.
>
> ### Q1: This paper only considers unstructured pruning. I suppose that the findings of pruning plasticity can also generalize to the structured setting. It would be nice to test pruning plasticity on the structured pruning.
>
> * We have added experiments on structured pruning with Li et al [1]’s filter pruning method. Due to the time limitation, we have only finished the uniform pruning experiments with ResNet-20 on CIFAR-10. Still, the results are highly consistent with the unstructured pruning. We will finish the rest of the experiments and add all of them in the next version.
>
> ### Q2: The authors should cite [2,3,4] for the zero-cost neurodegeneration.
>
> * Thanks for your suggestions, we will cite all of them in the next version.
>
> ### Q3: It is unclear to me if the pruned weights are still updated or not during the backward pass. This would cause a significant computational difference. It would be nice to make this point clearer.
>
> * We fully agree with you that it is important to keep the backward pass sparse as it would cause a large amount of overall computation. We clarify that the pruned weights are indeed forced to zero after and before each optimization update, and thus, the pruned weights are not updated during the backward pass. This point will be emphasized in the revision.
>
> ### Q4: I suggest the authors to put more emphasis on the dynamic sparse training part (GraNet-ST) rather than GraNet, which is more promising in terms of the accuracy-efficiency trade-off. The description of GraNet-ST in the current version is somehow unclear, especially for readers who are not familiar with sparse training.
>
> * We agree that the description of GraNet-ST can be clearer. We will add more elaborate descriptions in the revised/camera-ready paper. While the sparse-to-sparse method (GraNet-ST) is more promising to reach a better sweat spot between the performance and efficiency, what we want to highlight is the importance of consulting the information obtained during training, e.g., gradient and magnitude, to discover more accurate sparse subnetworks. The observations that recently proposed techniques for pruning before training still cannot match the performance of after-training pruning [2] also indicate that the information obtained during training is likely to be still necessary for finding sparse networks with sufficient/matching performance.
>
> [1]  [Li, Hao, et al. "Pruning filters for efficient convnets."](https://arxiv.org/abs/1608.08710)
>
> [2]  [Frankle, Jonathan, et al. "Pruning neural networks at initialization: Why are we missing the mark?."](https://arxiv.org/abs/2009.08576)

---

### Official Review · Reviewer_VuBd · 2021-07-18

**Rating:** 6
**Confidence:** 3

**Summary:**

This paper understands the during-training pruning from the perspective of training plasticity which is the ability of the pruned networks to recover the original performance. A technique named neuroregeneration is further shown to improve the training plasticity. Based on these findings, a gradual magnitude pruning (GMP) method with neuroregeneration (GraNet), and its dynamic sparse training variant are proposed. Experiments show the effectiveness of GradNet and GradNet-ST.

**Limitations And Societal Impact:**

The authors have addressed the limitations and potential negative societal impact of their work

**Main Review:**

Pros

The paper is well-written. The contributions are clearly presented.

Model compression is an important topic in scaling deep models on mobile devices. It is essential to design effective channel pruning method.

The key idea of neuron regeneration is well-motivated.

Extensive experiments are conducted to show the superiority of GradNet..

Cons

It seems that GradNet achieves marginal improvements over STR on the ImageNet dataset.

Does neuron regeneration work on a structured pruning framework?

The training plasticity is sensitive to hyper-parameters. It would be better to provide experiments to show how neuron regeneration helps training plasticity.


**Time Spent Reviewing:**

12 hours

---

> ### Author Response · Authors · 2021-08-10
> **Response to Reviewer VuBd:**
>
> We thank you very much for your positive comments and the time spent on our paper!
>
> ### Q1: It seems that GradNet achieves marginal improvements over STR on the ImageNet dataset.
>
> * We agree that GraNet is marginally better over STR on ImageNet. Meanwhile, we stress that it is only one of our goals to improve the performance of during-training pruning. As we emphasize in the introduction and at the beginning of Section 3, another more important goal that we wish to emphasize is the merit of pruning during training. Compared with the recently proposed works, i.e., LTH [1] and SNIP [2], during-training pruning is an efficient yet performant class of pruning methods that have received much less attention (agreed by Reviewer **RYno**).  We quantitatively study the effect of pruning throughout training from the perspective of pruning plasticity, helping to understand several separately reported phenomena in the literature.
>
> * Based on insights from pruning plasticity, we further modify an existing pruning-during training methods GMP and demonstrate the superiority of during-training pruning in terms of the accuracy-efficiency trade-off. Therefore, the most direct baseline of our method is GMP. Our results demonstrate that GraNet consistently outperforms GMP with various architectures on different datasets.
>
> * Moreover, our insight leads to the more competitive sparse-to-sparse method, GraNet-ST, which outperforms STR by 0.5% at a very challenging sparsity of 90%.
>
> ### Q2: Does neuron regeneration work on a structured pruning framework?
>
> * Yes, neuron regeneration also works on the structure pruning framework. Please see the overall response to all reviewers above for structured pruning experiments on pruning plasticity. The results are consistent with the reported results of the unstructured pruning.
>
> ### Q3: The training plasticity is sensitive to hyper-parameters. It would be better to provide experiments to show how neuron regeneration helps training plasticity.
>
> * Neuron regeneration is quite robust to hyperparameters. As you can see in the bottom row of each pruning plasticity figure, neuroregeneration significantly improves the pruning plasticity of all the cases, for models with different original sparsity and different pruning rates.
>
> * Additionally, as suggested by Reviewer **93rT**, we also test the sensitivity of GraNet-ST on the initial sparsity hyperparameter. GraNet-ST is quite robust to the initial sparsity and can consistently outperform RigL by over 1% at different initial sparsities. Please see our response to Reviewer **93rT**’s question “I couldn't see any ablations on the initial sparsity of GraNet-ST method. It would be nice to see the effect of initial sparsity.” for more details.
>
> [1] [Frankle, Jonathan, and Michael Carbin. "The lottery ticket hypothesis: Finding sparse, trainable neural networks.", ICLR 2019.](https://arxiv.org/abs/1803.03635)
>
> [2] [Lee, Namhoon, Thalaiyasingam Ajanthan, and Philip HS Torr. "Snip: Single-shot network pruning based on connection sensitivity.", ICLR 2019.](https://arxiv.org/abs/1810.02340)

---

### Author Response · Authors · 2021-08-10
**Response to all reviewers**

We would like to thank the reviewers and the area chairs for their time and insightful comments.

We are glad that they found our paper to be novel and interesting (Reviewer **93rT**), well-motivated (Reviewers **VuBd** & **RYno**), and to have done a good job on comparing a large set of methods (Reviewers **93rT** & **VuBd** & **RYno**). We are pleased that Reviewer **VuBd** finds that our method is solid and that Reviewer **93rT** considers our findings of pruning plasticity to be quite interesting. We begin by addressing some common comments:

 &nbsp;
 &nbsp;

\# **Pruning plasticity of structured pruning**  (Reviewers **VuBd** & **RYno**)

 &nbsp;
 &nbsp;

We have added the study of pruning plasticity on structured pruning. In particular, we choose the filter pruning method used in Li et al. [1] implemented with ResNet-20 on CIFAR-10. The pruning criterion is the absolute weight sum of each nonzero filter and the regeneration criterion is the absolute gradient sum of each zero filter. We first pre-train four sets of neural networks from scratch with various structured densities, including 100%, 90%, 50%, and 30%, noted as “Model Density” in the figure title. To measure the plasticity of these pre-trained models, we choose four different pruning rates noted as  “Pruning rate” to remove filters from these pre-trained models.

For convenience, we re-explain the definition of pruning plasticity here. Pruning plasticity is defined as $t_{Contrain^{k}(W_t \odot m_t, a_t)}-t_{pre}$, where $t_{pre}$ is the test accuracy measured before pruning and $t_{Contrain^{k}(W_t \odot m_t, a_t)}$ is the test accuracy measured after $k$ epoch of continued training $Contrain^{k}(W_t \odot m_t, a_t)$.  **The higher value means the better ability of the pruned model to recover the original performance.**

The results are averaged from 3 different runs shown as below.  The results are consistent with the reported results of the unstructured pruning. Overall, structured pruning is more vulnerable to higher sparsity pruning than unstructured pruning. The pruning plasticity of structured pruning also gradually decreases as the learning rate drops. Again, allowing new filters to regenerate significantly improves the pruning plasticity for models at all sparsity levels.

 &nbsp;

|  Model Density = 100% | Regeneration |  &nbsp;  epoch 20   |  &nbsp;  epoch 40  |   &nbsp; epoch 100  |  &nbsp; epoch 110 |  &nbsp;  epoch 140  | &nbsp;  epoch 150  |
| ------------- | :-----------: |:-----------:|:-------------:|:-------------:|:-------------:|:-------------:|:-------------:|
|                  |                 |lr = 0.1  |  lr = 0.1  |  lr = 0.01  |   lr = 0.01   |    lr = 0.001 | lr = 0.001|
| pruning rate=0.1 | No |  4.38$\pm$2.66  | 2.18$\pm$1.62  | -0.35$\pm$0.10  | 0.31$\pm$0.40  | -0.57$\pm$0.07 | -0.56$\pm$0.06 |
| pruning rate=0.1 | Yes | 4.23$\pm$2.14  | 2.46$\pm$1.24  | -0.29$\pm$0.04  |  0.45$\pm$0.52 | -0.41$\pm$0.11 | -0.33$\pm$0.07 |
| pruning rate=0.3 | No | -9.81$\pm$2.64 | -12.23$\pm$0.38 | -17.36$\pm$0.47  | -17.32$\pm$0.48 | -20.02$\pm$0.07 | -19.98$\pm$0.05|
| pruning rate=0.3 | Yes | 4.62$\pm$1.95 | 2.19$\pm$1.36 | -1.08$\pm$0.32 |0.09$\pm$0.23 | -1.17$\pm$0.06| -1.07$\pm$0.05|
| pruning rate=0.5 | No | -26.51$\pm$1.55| -28.83$\pm$1.67 | -36.23$\pm$0.27 | -34.97$\pm$0.71  | -37.85$\pm$0.56  | -37.79$\pm$0.41 |
| pruning rate=0.5 | Yes | 2.10$\pm$2.83 | 0.46$\pm$1.69 | -2.60$\pm$0.17 | -1.81$\pm$0.37  | -2.89$\pm$0.15| -2.74$\pm$0.04 |
| pruning rate=0.9 | No  |-59.99$\pm$2.29 | -62.54$\pm$0.94 | -72.60$\pm$0.32 | -71.37$\pm$0.49 | -76.04$\pm$0.95 | -76.51$\pm$1.34 |
| pruning rate=0.9 | Yes  | -7.73$\pm$3.39 | -9.65 $\pm$ 2.14 | -16.45 $\pm$ 1.26 | -15.70 $\pm$ 0.54 | -19.37 $\pm$ 0.41|-20.66 $\pm$ 0.57 |

 &nbsp;

|  Model Density = 90% | Regeneration |  &nbsp;  epoch 20   |  &nbsp;  epoch 40  |   &nbsp; epoch 100  |  &nbsp; epoch 110 |  &nbsp;  epoch 140  | &nbsp;  epoch 150  |
| ------------- | :-----------: |:-----------:|:-------------:|:-------------:|:-------------:|:-------------:|:-------------:|
|                  |                 |lr = 0.1  |  lr = 0.1  |  lr = 0.01  |   lr = 0.01   |    lr = 0.001 | lr = 0.001|
| pruning rate=0.1 | No |  6.47$\pm$2.36  | 1.23$\pm$2.00 | 0.00$\pm$0.18  | 0.24$\pm$0.35  |  -0.31$\pm$0.16 | -0.43$\pm$0.07 |
| pruning rate=0.1 | Yes | 3.99$\pm$0.33  | 0.97$\pm$0.87  | -0.10$\pm$0.02  | 0.26$\pm$0.07 | -0.23$\pm$0.25 | -0.33$\pm$0.10 |
| pruning rate=0.3 | No | -7.75$\pm$2.63 | -11.86$\pm$1.39 | -17.30$\pm$0.68 | -17.00$\pm$0.10 | -19.85$\pm$0.10 | -19.94$\pm$0.07 |
| pruning rate=0.3 | Yes| 3.92$\pm$0.49 | 0.49$\pm$0.06 | -0.78$\pm$0.01 | -0.74$\pm$0.10 | -1.04$\pm$0.07 | -1.23$\pm$0.05 |
| pruning rate=0.5 | No | -25.19$\pm$2.71 | -29.53$\pm$2.21 | -35.43$\pm$0.35 | -35.32$\pm$0.61  | -37.96$\pm$0.35  | -38.37$\pm$0.29 |
| pruning rate=0.5 | Yes | 2.31$\pm$0.41| -1.25$\pm$0.15 | -2.22$\pm$0.03 | -2.66$\pm$0.19  | -2.88$\pm$0.01 | -2.78$\pm$0.08 |
| pruning rate=0.9 | No | -57.97$\pm$2.21 | -62.55$\pm$1.83 | -72.22$\pm$0.33  | -71.82$\pm$0.13 | -78.32$\pm$0.68 | -78.45$\pm$0.34|
| pruning rate=0.9 | Yes | -7.10$\pm$1.36 | -12.90$\pm$1.20 | -18.34$\pm$0.47 | -18.83$\pm$0.67 | -21.12$\pm$1.02 | -24.74$\pm$3.33 |

 &nbsp;

|  Model Density = 50% | Regeneration |  &nbsp;  epoch 20   |  &nbsp;  epoch 40  |   &nbsp; epoch 100  |  &nbsp; epoch 110 |  &nbsp;  epoch 140  | &nbsp;  epoch 150  |
| ------------- | :-----------: |:-----------:|:-------------:|:-------------:|:-------------:|:-------------:|:-------------:|
|                  |                 |lr = 0.1  |  lr = 0.1  |  lr = 0.01  |   lr = 0.01   |    lr = 0.001 | lr = 0.001|
| pruning rate=0.1 | No |  -2.84$\pm$1.06  | -3.29$\pm$1.19 | -7.37$\pm$0.85  | -6.85$\pm$0.84  | -8.02$\pm$0.89 | -8.04$\pm$0.91 |
| pruning rate=0.1 | Yes | 4.12$\pm$0.91  | 4.74$\pm$1.83  | 0.63$\pm$0.98  | 1.41$\pm$0.35 | 0.37$\pm$1.50 | 0.30$\pm$0.70 |
| pruning rate=0.3 | No | -11.05$\pm$1.31 | -11.39$\pm$1.66 | -16.78$\pm$1.15  | -16.08$\pm$0.92 | -17.24$\pm$1.28 | -17.26$\pm$1.27 |
| pruning rate=0.3 | Yes| 3.69$\pm$3.04 | 3.52$\pm$2.62 | -0.80$\pm$2.41 | 0.72$\pm$0.58 | -0.23$\pm$1.89 | 0.21$\pm$1.02 |
| pruning rate=0.5 | No | -19.36$\pm$1.51 | -19.42$\pm$1.49 | -25.95$\pm$1.50 | -25.41$\pm$1.24  | -26.94$\pm$1.91 | -26.92$\pm$1.81 |
| pruning rate=0.5 | Yes | 3.38$\pm$3.40 | 3.48$\pm$2.39 | -1.68$\pm$3.03 | -0.96$\pm$1.63 | -2.26$\pm$1.91 | -1.69$\pm$1.13 |
| pruning rate=0.9 | No | -37.04$\pm$0.99 | -37.05$\pm$1.02  | -43.94$\pm$1.00  | -43.39$\pm$0.71 | -44.43$\pm$1.06 | -44.48$\pm$1.00|
| pruning rate=0.9 | Yes | -3.17$\pm$1.33 | -4.11$\pm$2.89 | -8.13$\pm$2.36 | -7.21$\pm$2.91 | -13.70$\pm$2.24 | -13.05$\pm$4.27 |

 &nbsp;


|  Model Density = 30% | Regeneration |  &nbsp;  epoch 20   |  &nbsp;  epoch 40  |   &nbsp; epoch 100  |  &nbsp; epoch 110 |  &nbsp;  epoch 140  | &nbsp;  epoch 150  |
| ------------- | :-----------: |:-----------:|:-------------:|:-------------:|:-------------:|:-------------:|:-------------:|
|                  |                 |lr = 0.1  |  lr = 0.1  |  lr = 0.01  |   lr = 0.01   |    lr = 0.001 | lr = 0.001|
| pruning rate=0.1 | No |  -4.37$\pm$0.66  | -6.12$\pm$0.56 | -7.5$\pm$1.26   | -7.21$\pm$1.04  |  -8.00$\pm$0.69 | -7.89$\pm$0.65   |
| pruning rate=0.1 | Yes | 3.89$\pm$0.26  | 1.54$\pm$0.07  | 1.12$\pm$1.57  | 1.17$\pm$0.66 | -1.09$\pm$0.91 | 0.62$\pm$1.36 |
| pruning rate=0.3 | No | -4.68$\pm$1.36 | -6.27$\pm$0.81 | -7.64$\pm$1.35  | -7.41$\pm$1.09  | -8.29$\pm$0.75 | -8.24$\pm$0.74 |
| pruning rate=0.3 | Yes | 3.45$\pm$0.32 | 2.11$\pm$0.41 | 0.69$\pm$1.44 | 0.90$\pm$0.76 | -1.43$\pm$0.95 | 0.39$\pm$1.45 |
| pruning rate=0.5 | No | -13.19$\pm$1.16 | -15.38$\pm$0.42 | -16.95$\pm$1.26 | -16.76$\pm$0.85   | -17.67$\pm$0.77 | -17.54$\pm$0.69 |
| pruning rate=0.5 | Yes | 2.42$\pm$0.43 | 0.79$\pm$0.39 | -1.44$\pm$1.36 | 0.13$\pm$1.30 | -2.43$\pm$0.15 | -1.33$\pm$1.55 |
| pruning rate=0.9 | No | -22.38$\pm$0.90 | -24.18$\pm$0.37  | -26.40$\pm$1.15  | -26.17$\pm$0.87 | --26.87$\pm$0.51 | -26.78$\pm$0.49 |
| pruning rate=0.9 | Yes | 0.51$\pm$0.35 | -2.08$\pm$0.52 | -5.81$\pm$0.32 | -1.42$\pm$0.96 | -5.85$\pm$1.50 | -5.84$\pm$0.98 |

Due to the limited time, we have only finished the experiments for ResNet-20 with uniform pruning. We will finish the rest of the unstructured experiments and add all of them in the next version of the paper in an appendix.

[1]  [Li, Hao, et al. "Pruning filters for efficient convnets." ](https://arxiv.org/abs/1608.08710)

---

### Decision · Program_Chairs · 2021-09-27

**Decision:**

Accept (Poster)

**Comment:**

This paper studies how to optimize the performance of sparse models which are pruned during the training process by combining during-training pruning with the growth criterion originally proposed in RigL. All reviewers felt the paper was clear and covered a timely and important topic, though there were some concerns regarding clarity and comparison to prior work which were resolved through the author discussion. As such, I recommend the paper be accepted.